# Evidence of a causal and modifiable relationship between kidney function and circulating trimethylamine *N*-oxide

The host-microbiota co-metabolite trimethylamine *N*-oxide (TMAO) is linked to increased cardiovascular risk but how its circulating levels are regulated remains unclear. We applied "explainable" machine learning, univariate, multivariate and mediation analyses of fasting plasma TMAO concentration and a multitude of phenotypes in 1,741 adult Europeans of the MetaCardis study. Here we show that next to age, kidney function is the primary variable predicting circulating TMAO, with microbiota composition and diet playing minor, albeit significant, roles. Mediation analysis suggests a causal relationship between TMAO and kidney function that we corroborate in preclinical models where TMAO exposure increases kidney scarring. Consistent with our findings, patients receiving glucose-lowering drugs with reno-protective properties have significantly lower circulating TMAO when compared to propensity-score matched control individuals. Our analyses uncover a bidirectional relationship between kidney function and TMAO that can potentially be modified by reno-protective anti-diabetic drugs and suggest a clinically actionable intervention for decreasing TMAO-associated excess cardiovascular risk.

Over the past two decades, the central role of the commensal gut microbiota in pathologies such as atherosclerosis and type-2 diabetes (T2D) has gained prominence[1]. The microbiota can influence host pathophysiology by producing molecules that directly alter metabolism and/or modulate cellular signaling either locally in the gut or systemically via the circulation[1,2].

Trimethylamine *N*-oxide (TMAO) is the phase-one liver *N*-oxide of trimethylamine (TMA). TMA is a product of the microbial[3–6] (predominately Firmicutes) metabolism of phosphatidylcholine[7,8], choline[8], and *L*-carnitine[9–11], components of the high-fat, high red meat western diet. TMA is taken up from the gut via the hepatic portal vein and *N*-oxidized into TMAO by host flavin mono-oxygenase 3[12]. High circulating levels of TMAO have been linked to increased thrombotic and cardiovascular risk in animal and human studies, even after adjustment for known cardiovascular risk factors[7–9,13]. TMAO is therefore proposed to mediate the higher cardiovascular risk associated with high red meat and fat intake[14]. Fish, the consumption of which is

associated with reduced incidence of cardiovascular disease[14], is also a rich source of TMAO[15]. Dietary TMAO is subject to retro-conversion: i.e., it can undergo microbial reduction to TMA by *Enterobacteriaceae* followed by hepatic conversion back to TMAO[16]. Concurrent with diet and microbiota composition, TMAO plasma levels also reflect age[17], sex[18], kidney function[19–21], and chronic diseases[7,22]. To date, the relative contribution of each of these factors to circulating TMAO levels and, therefore, elevated cardiovascular risk remains unclear. Understanding how serum TMAO levels are regulated could uncover host TMAO mechanistic targets and identify modifiable and actionable therapeutic factors to lower circulating TMAO levels.

Here, by using a data-driven "explainable" machine learning (ML) strategy[23], multivariate and univariate analyses of epidemiological data and mechanistic studies in cultured cells and rodents, we sought to objectively identify variables influencing serum TMAO levels in participants of the European multicenter MetaCardis study. Moreover, by taking advantage of the unique cross-sectional MetaCardis design, we

✉ e-mail: p.andrikopoulos04@imperial.ac.uk; karine.clement@inserm.fr; m.dumas@imperial.ac.uk

queried how variables influencing circulating TMAO manifest at different stages of cardiometabolic disease. Capitalizing on the ML analysis, we aimed to identify (i) novel host TMAO-related mechanistic targets and (ii) a rationale for future interventions that could reduce circulating TMAO levels and thereby decrease associated excess cardiovascular risk.

In epidemiological studies, we observed that kidney function is the main modifiable factor consistently regulating fasting serum TMAO levels, and our preclinical studies align with the suggestion that elevated circulating TMAO adversely affects kidney function, by increasing kidney fibrotic injury. Further supporting the strong interplay between kidney function and fasting circulating TMAO, patients with T2D in the cohort prescribed new-generation anti-diabetics (GLP-1 Receptor Agonists[24]; GLP-1RAs) with evidenced reno-protective effects[25] had lower serum circulating TMAO levels when compared to propensity-score matched controls (Fig. 1).

## Results

In the first analysis, we investigated how TMAO levels change depending on disease classification in the MetaCardis population. We confirmed, circulating TMAO significantly increased with cardiometabolic disease severity, in line with previous reports[7,22] (Supplementary Fig. 1A). To identify determinants of circulating TMAO manifesting at prodromal stages of cardiometabolic disease, we focused on the subset of the MetaCardis cohort termed MetaCardis Body Mass Index Spectrum subset (BMIS[26]; $N = 837$) comprising obese/overweight individuals presenting with a range of metabolic syndrome features but not overt T2D or ischemic heart disease (IHD) (Supplementary Table 1).

### Fasting serum concentration of TMAO is associated with worse cardiometabolic profiles in BMIS MetaCardis population

We explored how TMAO correlated with bioclinical variables related to cardiometabolic health in BMIS individuals. In this group, circulating TMAO was associated with reduced values of estimated glomerular filtration rate (eGFR, Spearman rho = −0.124, pFDR = 0.06) and higher fasting concentrations of uric acid (rho = 0.114, pFDR = 0.09) after adjustment for age, sex, country of recruitment (demographics thereafter) and body mass index (BMI) (Supplementary Fig. 1B). Higher TMAO also positively associated with indicators of central adiposity, including BMI (rho = 0.107, pFDR = 0.09), visceral body fat rating (rho = 0.122, pFDR = 0.09) and waist circumference (rho = 0.119, pFDR = 0.09), after demographics adjustment (Supplementary Fig. 1C). Moreover, BMIS individuals with hypertension (systolic blood pressure over 140 mmHg, or receiving therapy for high blood pressure; "Methods") had higher plasma TMAO ($P = 0.01$, Mann–Whitney test; Supplementary Fig. 1D). In line with previous studies[7,20], objectively dividing BMIS participants into TMAO clusters using k-means[27] revealed that those in the cluster with the highest TMAO levels had consistently worse cardiometabolic traits and were significantly older when compared to those in the cluster with lowest TMAO levels (Supplementary Fig. 1E and Supplementary Table 2). Traits of cardiometabolic risk included altered eGFR, elevated liver enzymes, and systolic blood pressure (Supplementary Fig. 1F–I).

### Age and altered kidney function variables are the main drivers of circulating TMAO in BMIS

To better understand which variables (Supplemental Data 1) affect circulating TMAO most, we trained extreme gradient-boosted decision-tree models. We used fivefold cross-validation to predict the Explained Variance (EV) of each variable group on plasma TMAO in the left-out BMIS participants after 100 iterations (Fig. 2A). Microbiota composition alone performed poorly (EV 2%) whilst diet, another purported major contributor to TMAO production, explained less than 5% of TMAO variance. Serum metabolomics, excluding TMAO and its

precursor TMA, was the best predictor (EV 12%) with demographics second, explaining 10% of circulating TMAO variance. [1]H-NMR urine metabolomics, excluding TMA and dimethylamine, was the worst predictor explaining 1.5% of TMAO variance and correcting for urinary creatinine, computed by [1]H-NMR ("Methods"), did not improve predictions explaining 1% of TMAO variance on average (Supplementary Fig. 2A). The full model, containing all variable groups, accounted, on average, for 18.4% of TMAO variability. The average predicted TMAO values by the full model significantly correlated with measured TMAO values (rho = 0.473, $P < 2.2 \times 10^{-16}$; Supplementary Fig. 2B).

We next assessed the independent contribution of each variable group to the predictive power of the full model by training algorithms as above but removing one feature group at a time for 100 iterations. Almost 40% of the explainable variance of the full model (set to 100%) was contributed by serum metabolomic variables, with biological, dietary and microbiota taxonomic variables adding 5.5%, 4%, and 3.4%, respectively, independently explained variance (Fig. 2B). Other variable categories displayed negligible contribution to prediction suggesting considerable information overlap with the metabolomic, metagenomic, biological and dietary datasets.

Using feature attribution analysis (SHapley Additive exPlanations; SHAP[23,28]), we assessed how individual variables drive TMAO-predicting models. SHAP analysis identified 24 variables that contributed more than 4% of the regularized TMAO standard deviation (SD) to model outcomes. Of those, age affected predictions the most, followed by eGFR, urinary betaine, percentage of visceral body fat and serum butyryl-carnitine (Fig. 2C). Besides eGFR, additional variables associated with kidney function including plasma urea[29], the uremic toxin $p$-cresol[30] or markers of kidney function decline (i.e., serum albumin[29]), were among those affecting model outcomes the most. This analysis suggests that kidney function is a major determinant of circulating TMAO. The impact of eGFR on model outcomes for BMIS individuals was bimodal with values over 90 mL/min/1.73 m², the clinical cut-off value for normal kidney function in adults[25], predicting reduced plasma TMAO and lower values resulting in increased predicted circulating TMAO (Fig. 2D).

We next trained algorithms predicting TMAO using the 24 variables identified by our SHAP analysis ("top SHAP" model) and compared it to models trained by traditional clinical risk factors[31] or the full model (Fig. 2E). The "top SHAP" model significantly ($P < 2.2 \times 10^{-16}$, Mann–Whitney test) improved predictions when compared to the full model (EV 21% vs. 18%, on average), presumably by removing noise, supporting the importance of the variables identified by the SHAP analysis.

To confirm that tree-based ML models are the most appropriate for our analysis we also built Least Absolute Shrinkage and Selection Operator (LASSO) models to predict circulating TMAO in the left-out group using again fivefold cross-validation with all the available variables (full model) as input in BMIS ($N = 582$; "Methods"). LASSO explained on average 14.9% of circulating TMAO variance after 100 iterations (Supplementary Fig. 3A; see source data for lambda and $R^2$ values of each iteration) as opposed to 18.4% by boosted trees for the full model (Fig. 2A). This analysis supports the appropriateness of tree-based ML models for predicting circulating TMAO in our population.

To determine how much of the variances of TMAO and of the other metabolites most strongly associated with its levels in our ML models (Fig. 2C; butyryl-carnitine, betaine, $p$-cresol and betaine_U, oxaloacetate_U) is explained by eGFR we built linear-regression models with each metabolite as the dependent variable and eGFR as the independent variable. Kidney function explained 7% of TMAO variance in BMIS (Fig. 2F; Pearson's $r = -0.26$, $P = 5.4 \times 10^{-14}$, $N = 837$) whilst the explained variance for the other metabolites ranged from 6% to 1.4% for $p$-cresol and urinary oxaloacetic acid, respectively (Supplementary Fig. 4A–E). To further assess the varying relationship of metabolites with eGFR we computed boosted trees models predicting eGFR with

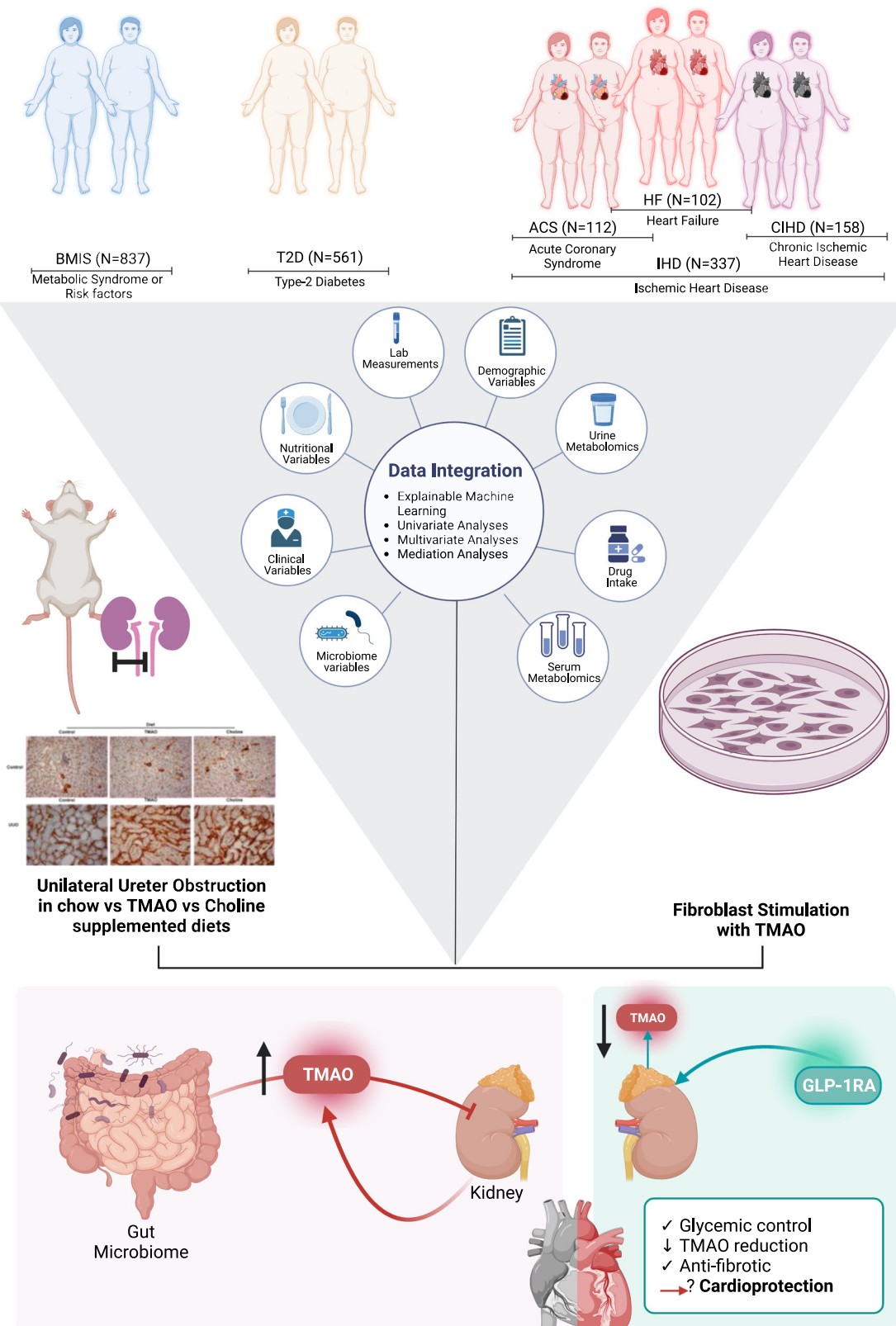

**Fig. 1 | Overview of study design and main findings.** Here we used an integrated approach comprising Machine Learning (ML), multivariate, univariate, and mediation analyses to objectively characterize host parameters contributing to plasma TMAO levels in the multicenter European MetaCardis study. We observed that kidney function is the main modifiable factor consistently regulating fasting serum TMAO levels (Figs. 2–4) and corroborated our epidemiological findings in preclinical models where TMAO increased kidney scarring (Fig. 5). Further supporting the strong interplay between kidney function and fasting circulating TMAO, patients with T2D in the cohort prescribed new-generation anti-diabetics (GLP-1 Receptor Agonists; GLP-1RAs) with reno-protective effects had lower serum circulating TMAO levels when compared to propensity-score matched controls (Fig. 6). Created with BioRender.com.

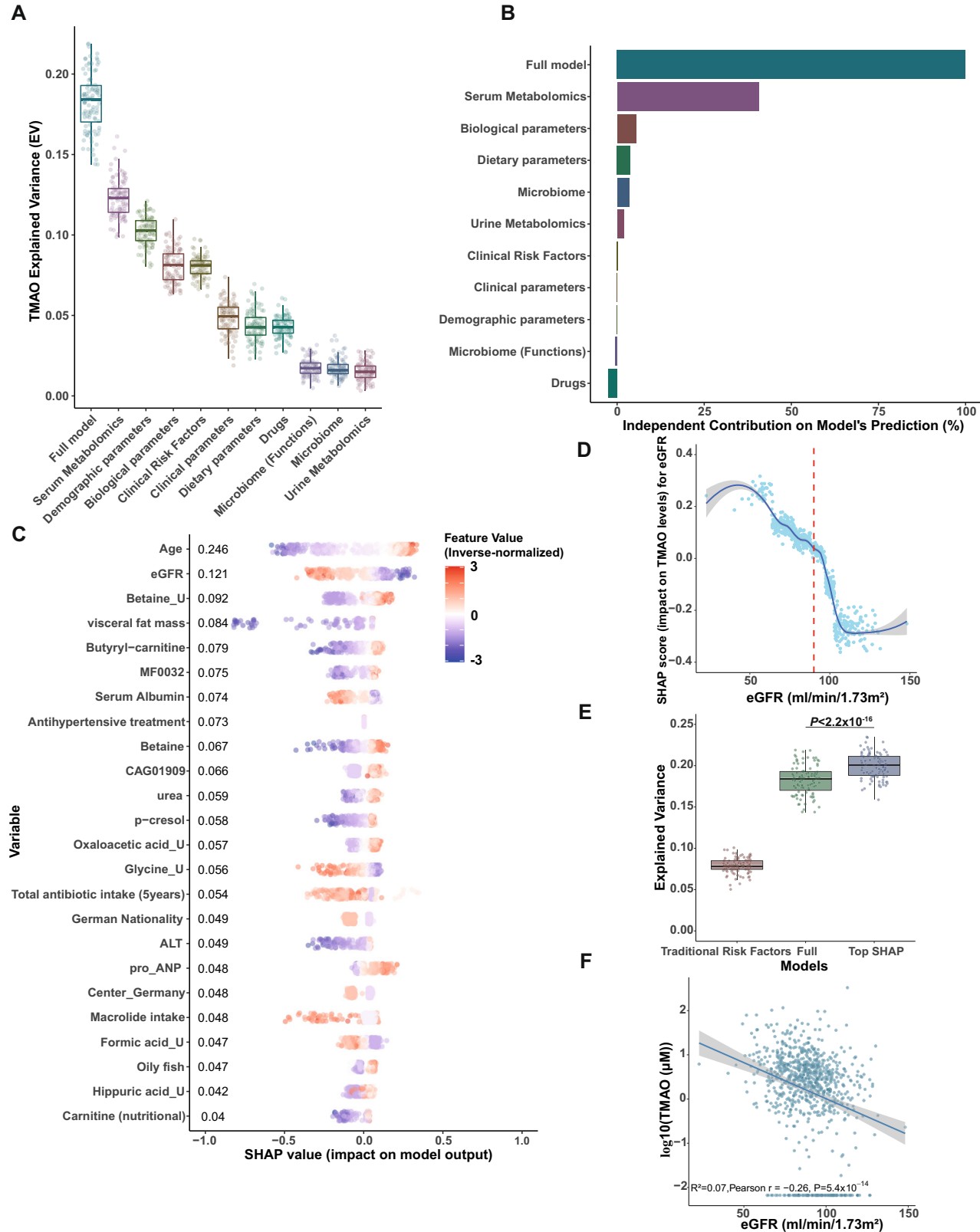

serum metabolomics as the input variable, with a similar methodology to TMAO. Serum metabolomics predicted on average 25% of eGFR variance after 100 iterations. From all the metabolites in our analysis TMAO was the top microbiota-derived compound that affected most strongly eGFR predictions (Supplementary Fig. 4F) in line with its reported exclusive glomerular secretion[32]. Collectively this analysis

suggests that in our population, TMAO is strongly interlinked with eGFR.

Several reports have previously highlighted the inverse correlation between TMAO and kidney function mostly in patients with Chronic Kidney Disease (CKD)[10,19,20,32–34], but there is limited evidence for the predominance of this relationship in the non-clinical range. The

**Fig. 2 | Age and parameters associated with kidney function are the main drivers of circulating TMAO in BMIS MetaCardis participants. A** Coefficients of determination (Explained Variance; EV) of predicted circulating TMAO levels determined by xgboost algorithms after fivefold cross-validation in the left-out group (Supplementary Table 3 for *n* numbers and xgboost parameters), trained exclusively on variables from each feature category (Supplemental Data 1 for variables included in each group), or the full model (all variables), after 100 iterations. **B** Averaged independent predictive contribution of each feature category to full model predictions of plasma TMAO, trained as in (**A**), calculated as the average reduction of EV achieved in relation to the full model (set to 100%) after removing all variables belonging to each feature group after 100 iterations. **C** Swarm plots of impact on model output (SHAP values) for each BMIS individual with complete phenotypic data (*N* = 582) for all variables contributing more than 4% to model predictions of regularized TMAO standard deviation, as determined by xgboost algorithms trained on each feature category. Mean absolute SHAP values from all BMIS participants (*N* = 582) are shown (in descending order) next to each variable. Individual dots, representing each participant, are colored by the inverse-

normalized value of the corresponding variable. U denotes urinary metabolites. **D** Dependance plot of eGFR values (*x* axis) versus their impact on model outcome (*y* axis) calculated for each individual in BMIS (*N* = 837) from algorithms trained on bioclinical variables, vertical red line indicates 90 mL/min/1.73 m$^2$. The curve was drawn using locally weighted scatterplot smoothing (LOWESS) and the shaded area indicates 95% confidence interval (CI). **E** Boxplots depicting EV ($R^2$) of circulating TMAO for BMIS participants computed from algorithms trained on clinical risk factors[29], the full model or all 24 variables contributing more than 4% of regularized TMAO standard deviation to predictions, as determined by SHAP analysis, after 100 iterations. Significance was determined by the two-sided Matt–Whitney test. **F** Linear-regression-based scatterplot showing correlation between serum TMAO (log-transformed) and estimated Glomerular Filtration Rate (eGFR, ml/min/1.73 m$^2$). Insert; unadjusted Pearson's *r*, *P* value and explained variance ($R^2$). Shaded area indicates 95% CI. **A**, **E** Center lines denote medians, box limits indicate the 25th and 75th percentiles, whiskers extend to the minimal and maximal values. Source data are provided as a Source Data file.

novelty of the present study includes ranking of a multitude of factors contributing to circulating TMAO levels and identification of kidney function as the top modifiable factor in a non-CKD population across the cardiometabolic disease spectrum.

## Diet and microbiota composition have a modest impact on fasting serum TMAO levels in BMIS individuals

Our ML models suggested that diet plays a modest role in determining circulating TMAO. To corroborate this, we assessed how consumption of food items rich in TMAO dietary precursors[3] affected circulating TMAO in BMIS participants with a Food Frequency Questionnaire[35] (FFQ; *N* = 763). After correcting for demographics and BMI, with the exception of oily fish (rho = 0.125, pFDR = 0.03), circulating TMAO was neither significantly associated with habitual consumption of red meat, eggs or milk nor with the estimated dietary intake of the micronutrients choline, carnitine or betaine, broadly in agreement with a previous study[3] (Fig. 3A). In addition, no significant correlations were found between consumption of these food items and the serum levels of TMAO precursors choline, betaine or γ-butyrobetaine[36]. Conversely, circulating TMAO positively correlated with plasma choline, betaine and γ-butyrobetaine, implying an association between the serum concentrations of TMAO precursors. Our findings that diet and particularly meat consumption does not associate with increased TMAO levels in our population do not contradict a number of well-designed human interventional trials[9–11] that have established a clear link between meat intake or *L*-carnitine supplementation and TMAO circulating levels. Instead, collectively, our analyses suggest that in non-interventional settings where most individuals consume meat daily (75–233 g/day for European adults[37]), clearance by the kidney and not dietary intake of TMAO precursors is the major determinant of fasting circulating TMAO levels and therefore of the excess cardiovascular risk associated with elevated TMAO (vegetarians and strict vegans aside, who are infrequent in the MetaCardis cohort with only 65/1741 participants reporting no red meat consumption).

To identify putative microbial taxa influencing serum plasma TMAO, we performed multivariate and univariate analyses. Principal coordinates analysis of Bray–Curtis dissimilarity matrices at the species level for BMIS individuals stratified into TMAO clusters revealed a significant difference in microbiota composition between clusters (*P* = 0.033; Fig. 3B). Consistent with previous reports[3,38] and our ML models, multivariate PERMANOVA analysis uncovered a significant, albeit weak, association between circulating TMAO and microbiota composition (*P* = 0.001; $R^2$ = 0.008, Fig. 3B) after demographics adjustment. TMAO levels significantly associated with the quantitative abundance of 65 bacterial species (corrected for bacterial load) in BMIS individuals after correcting for demographics and BMI, primarily (44/65) of the Firmicutes phylum (Fig. 3C). In agreement with Li et al.[3],

we did not identify any overlap between bacteria associated with circulating TMAO and higher red meat, milk or egg consumption in BMIS participants (*N* = 761; Fig. 3C). Contrasting low and high TMAO clusters revealed 215 differentially abundant bacterial species between these two groups (Fig. 3D; Mann–Whitney test).

We further analyzed the impact on circulating TMAO of the only species, an unknown bacterium taxonomically closely related to ruminoccoci (CAG01909), that contributed at least 4% of the TMAO SD in our prediction models and was significant (pFDR <0.05) for both correlation and differential abundance analyses (Fig. 3E). CAG01909 was more abundant and prevalent in the high TMAO cluster when compared to the low (Supplementary Fig. 5A, B). Finally, BMIS individuals harboring CAG01909 had significantly higher circulating TMAO than those that lacked it (Supplementary Fig. 5C), the CAG1909 abundance significantly correlating with TMAO concentration (Supplementary Fig. 5D).

This analysis corroborated the ML models suggesting a small, albeit significant, contribution of microbiota variations on circulating TMAO and identified a bacterium newly associated with higher serum TMAO in BMIS.

## Signatures predicting circulating TMAO shift in different disease groups

To identify putative common variables driving circulating TMAO levels in different disease groups, we trained ML algorithms for T2D (*N* = 561) and IHD sub-cohorts (*N* = 356). Similar to BMIS, microbiota composition and dietary variables alone performed poorly, explaining on average 2% and 1% of TMAO variance, respectively, in T2D and IHD individuals (Fig. 4A and Supplementary Fig. 6A, respectively). For the T2D cohort, serum metabolomics was again the best predictor (EV 12.8%) followed by bioclinical (EV 9.6%) and demographic variables (EV 8.7%) with the full model accounting for 16.2% of TMAO variance. For IHD, individual feature categories—except for serum metabolomics (EV 19.5%)—performed poorly, with the full model accounting for 16.1% of circulating TMAO variance (Supplementary Fig. 6A), possibly reflecting reduced power due to smaller sample size and/or IHD group heterogeneity. Feature attribution analysis revealed 19 variables in the T2D cohort and 10 variables in the IHD cohort that contributed more than 4% of regularized TMAO SD to model outcome (Fig. 4B and Supplementary Fig. 6B, respectively). In patients with T2D, model outcome was mostly affected by eGFR followed by age and the serum concentrations of the uremic toxin *p*-cresol[30] and *D*-threitol, also indicative of kidney function[39]. In patients with IHD, serum butyryl-carnitine, followed by age, alternative healthy eating score (aHEI) related to the ratio of white to red meat intake and the levels of the proinflammatory cytokine IP10, a marker of adverse cardiac remodeling[40] were the top variables. As in BMIS, the "top SHAP"

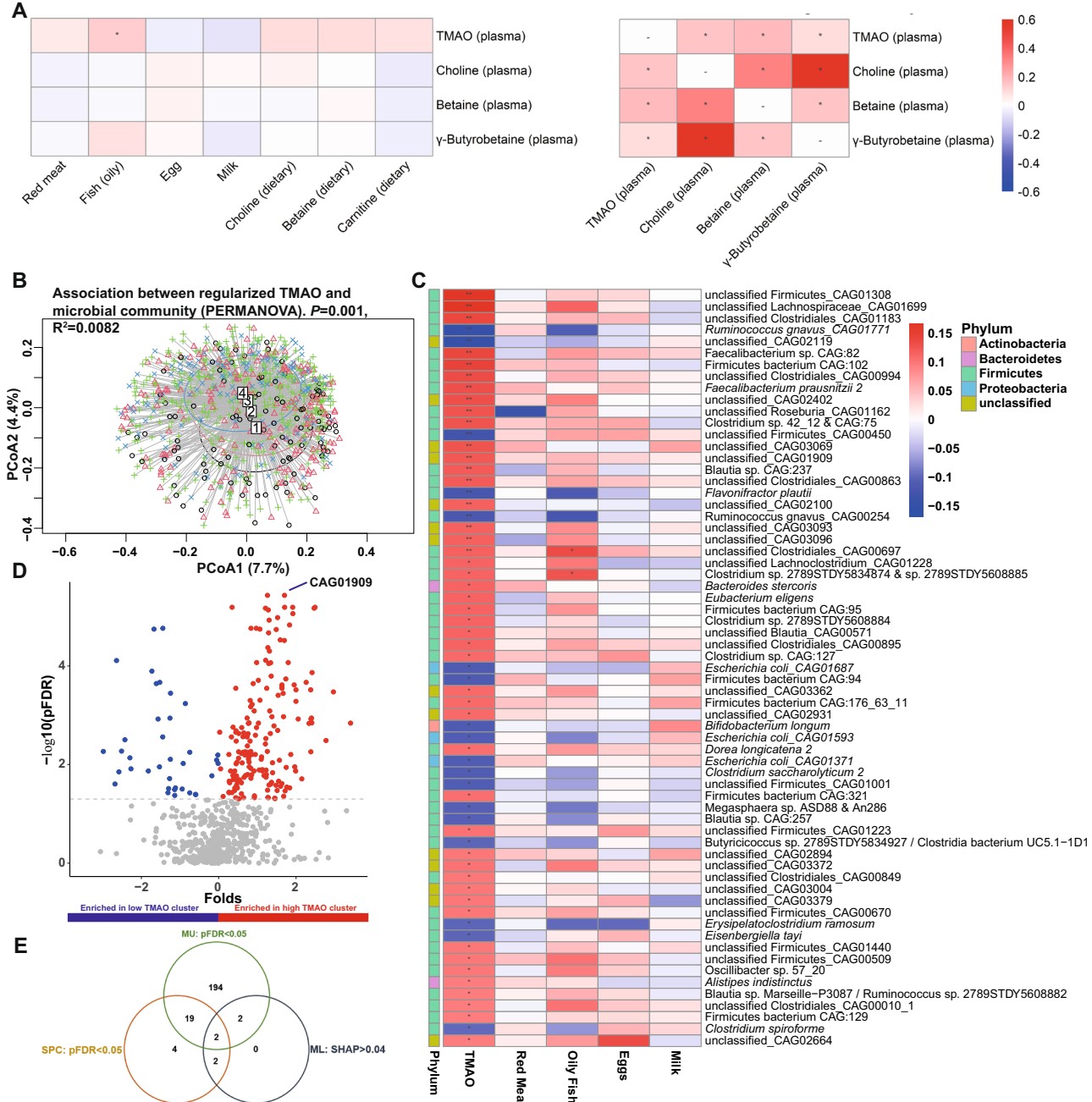

**Fig. 3 | Modest impact of diet and microbiota composition on circulating TMAO in BMIS MetaCardis subjects. A** Associations between circulating TMAO and its precursors with habitual consumption of food items rich in TMAO precursors (*N* = 763; left panel) or with its precursors themselves (right panel). **B** Principal coordinates analysis of Bray−Curtis dissimilarity matrices of participants (*N* = 834) stratified in TMAO clusters by the k-means algorithm (1 the lowest, 4 the highest) at the species level (input; 699 species present in at least 20% of the BMIS population). Insert; PERMANOVA (999 iterations) of taxonomic Bray−Curtis dissimilarity matrices association with regularized TMAO levels with age, sex, and country of recruitment as covariates. **C** Overlap of microbiome taxa significantly associated with circulating TMAO (Spearman partial correlations adjusted for age, sex country of recruitment and BMI) and the consumption of food items rich in TMAO precursors in BMIS participants (*N* = 763). **D** Volcano plot of differential bacterial species abundances between BMIS participants in the lowest (*N* = 101) and highest (*N* = 147) TMAO clusters (blue; taxa significantly depleted, red; taxa significantly enriched in the high TMAO cluster respectively, two-sided Mann−Whitney *U* test, pFDR<0.05). **E** Venn diagram summarizing the overlap between taxa associating with circulating TMAO according to our three complimentary analyses (SPC Spearman correlations, ML machine learning and feature attribution analysis; MU: two-sided Mann−Whitney *U* test between high and low TMAO clusters). For all *pFDR <0.05, **pFDR <0. 0.01. Source data are provided as a Source Data file.

models significantly improved predictions (EV 24% versus 16.2% for T2D; Fig. 4C and EV 21.3% versus 16.1% for IHD; Supplementary Fig. 6C) when compared to models trained with clinical risk factors or the full model.

Eight variables contributing more than 4% of TMAO SD were shared between BMIS and T2D, including, eGFR, age, and *p*-cresol

whilst there were six common features between BMIS and IHD (including eGFR, urea, age and butyryl-carnitine). Only three variables (eGFR, age and butyryl-carnitine) strongly contributed as predictors across all three disease groups (Fig. 4D).

We thus identified age and kidney function as the prominent variables influencing fasting circulating TMAO levels. To further

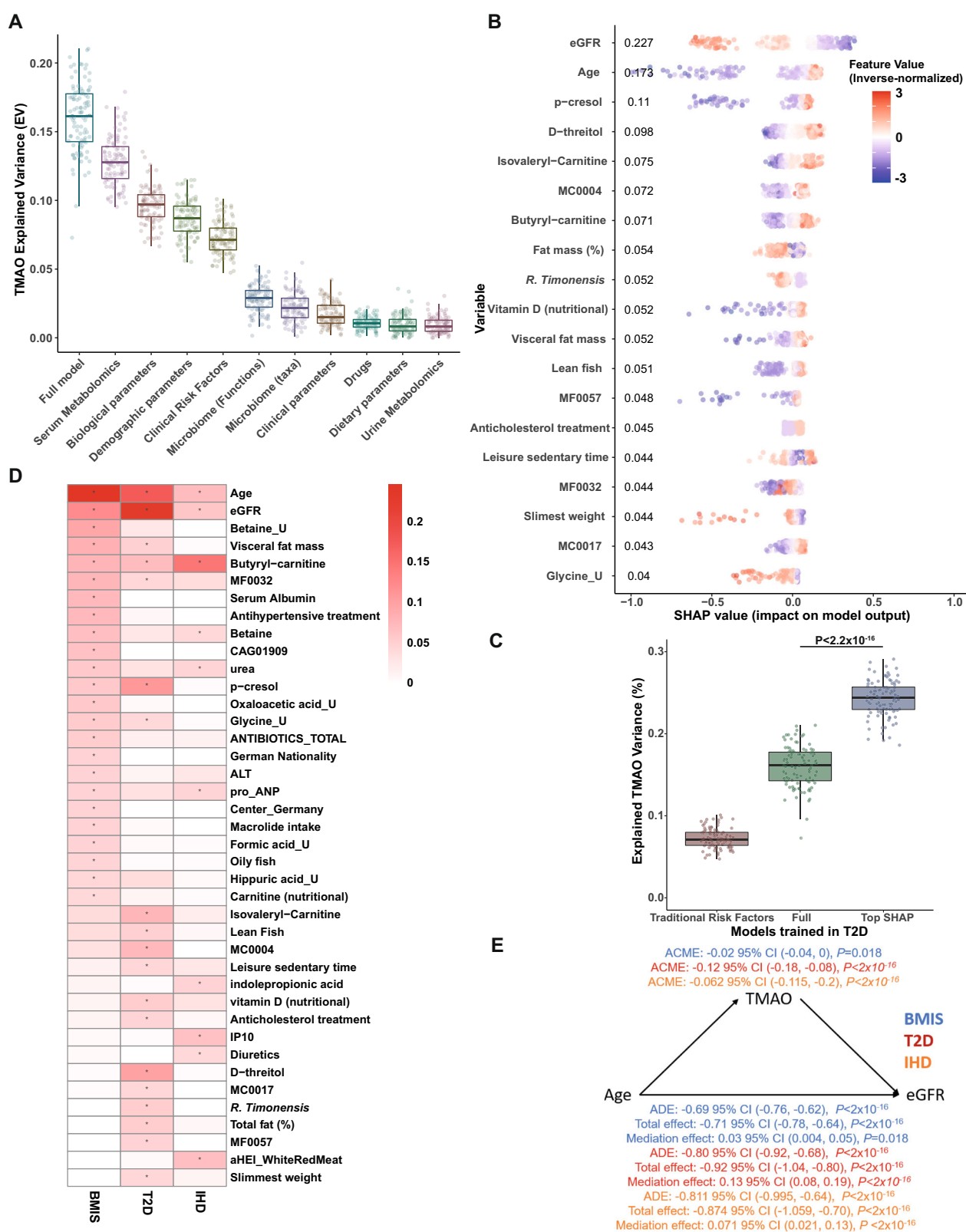

substantiate this relationship, we evaluated whether data fitted a model where eGFR causally mediates the increase of TMAO concentration with age. Under this model, kidney function achieved significance as mediating ~20% of the positive association between age and circulating TMAO levels in BMIS with its impact further strengthening in the T2D (55%) and IHD (51%) disease groups (Supplementary Fig. 6D), thus corroborating the ML analysis.

## Microbiota composition modestly affects plasma TMAO levels in T2D

In patients with T2D, ML and SHAP analyses revealed an inverse association between fecal *Romboutsia timonensis* quantitative abundance (corrected for bacterial load) and circulating TMAO (Fig. 4B), whilst no taxa strongly influenced TMAO predictions in the IHD subcohort (Supplementary Fig. 6B). *R. timonensis* inversely associated

**Fig. 4 | Signatures predicting circulating TMAO shift in different disease groups and TMAO causally mediates eGFR decline with age. A** Explained Variance (EV) of predicted serum TMAO levels determined by boosted decision trees (Supplementary Table 4 for *N* numbers and optimized xgboost parameters per variable group), trained exclusively on variables from each variable category (Supplemental Data 1 for a list of variables included in each group), or the full model (containing all variables), after 100 iterations in T2D MetaCardis patients. **B** Swarm plots of SHAP values (impact on model outcome) for each T2D MetaCardis participant with complete phenotypic data (*N* = 387); represented by individual dots, for all variables contributing more than 4% to model predictions of regularized TMAO standard deviation, computed from xgboost algorithms trained on each feature category. Numbers denote mean absolute SHAP values from all T2D participants (in descending order) next to their corresponding variable. Dots are colored by the inverse-normalized value of their corresponding variable. **C** Boxplots depicting Explained Variance (EV; $R^2$) of circulating TMAO in T2D individuals calculated by

algorithms trained on clinical risk factors[31], the full model containing all variables or all the variables contributing more than 4% of regularized TMAO standard deviation to T2D model predictions, as determined by SHAP analysis, after 100 iterations. **D** Heatmap depicting all the variables contributing at least 4% of regularized TMAO standard deviation in model predictions as determined by SHAP analysis in at least one of the MetaCardis disease groups. *Mean absolute SHAP value > 0.04. **E** Mediation analysis (see "Methods") computing the direct effect of TMAO on eGFR decline with age in BMIS (blue), T2D (red) or IHD (orange) MetaCardis participants. ADE: Average direct effect (of age on eGFR); ACME: average causal mediated effect (of TMAO on eGFR); Total effect: (cumulative effect of age and TMAO on eGFR (ADE + ACME)); Mediation effect: (% of the effect of age on eGFR attributed to TMAO). **A**, **C** Center lines denote medians, box limits indicate the 25th and 75th percentiles, whiskers extend to the minimal and maximal values. Source data are provided as a Source Data file.

with TMAO concentration (rho = −0.140, *P* = 0.0009; Supplementary Fig. 7A), T2D individuals with detectable *R. timonensis* had significantly lower TMAO levels (Supplementary Fig. 7B) and *R. timonensis* was depleted in T2D compared to BMIS participants (Supplementary Fig. 7C), in line with circulating TMAO (Fig. 1A). Conversely, in T2D and IHD CAG01909 abundance or presence, unlike BMIS, did not associate with higher circulating TMAO (Supplementary Figs. 7D–5F). Further investigation revealed that CAG01909 abundance was inversely associated with metformin intake (rho = −0.141, pFDR=0.042; Supplementary Fig. 7G), in accordance with the well-documented effect of drugs on the gut microbiome[41,42]. Irrespective of TMAO associations with individual species, similarly to BMIS multivariate PERMANOVA analysis uncovered a significant, albeit weak, association between circulating TMAO and microbiota composition (*P* = 0.001; $R^2$ = 0.005) after adjustment for demographics.

Collectively, these analyses suggest that the composition of the gut microbiota may only modestly influence circulating levels of TMAO in patients with T2D, and in subjects with IHD or T2D abundance of the CAG01909 taxon is not linked with serum TMAO.

### Inference analysis suggests that TMAO may causally mediate the decline of eGFR with age

ML analysis identified age and kidney function as the most prominent variables influencing TMAO. Therefore, we tested the inverse relationship: i.e., whether TMAO may mediate eGFR decline with age[43]. In BMIS, TMAO modestly, albeit significantly, modulated the inverse relationship between age and eGFR (mediation effect 3%, *P* = 0.018), whilst in T2D and IHD the impact of TMAO on kidney function decline with age strengthened (mediation effect 13%, $P < 2.2 \times 10^{-16}$ and 7.1%, $P < 2.2 \times 10^{-16}$; Fig. 4E). Mediation analysis thus indicated that TMAO, far from being a bystander, directly and adversely affects kidney function. Our finding is in agreement with a prospective study reporting that baseline TMAO positively associated with rates of eGFR decline[21] and with studies in animal models of CKD where TMAO diet supplementation increased kidney injury[44,45] and reduced eGFR.

Interestingly, mediation analysis showed that TMAO's adverse effect on kidney function strengthened at more severe stages of cardiometabolic disease, implying TMAO synergy with existing pathology (thereby providing "2-hit model"). Accordingly, we next employed preclinical models of kidney injury to gain mechanistic insights into the potential interplay between TMAO and kidney function and the putative molecular nature of the 2nd hit.

### TMAO increases trans-differentiation of human primary renal fibroblasts into myofibroblasts in conjunction with TGF-β1 signaling

Based on TMAO's detrimental association with kidney damage of diverse aetiologies[19,20] similar to renal fibrosis[46], we investigated the impact of TMAO on human primary renal fibroblasts (HRFs). In HRFs,

unlike platelets[47], TMAO stimulation resulted in rapid $[Ca^{2+}]_i$ increase (Fig. 5A). In endothelial cells, the ERK1/2 pathway is activated in a $Ca^{2+}$-dependent manner[48,49] and ERK1/2 activation exacerbates renal fibrosis[50]. Therefore, we investigated whether a similar pathway operates in HRFs. TMAO challenge increased phospho-ERK1/2 levels in a time- (Fig. 5B) and dose-dependent (Fig. 5C and Supplementary Fig. 8A) manner, at concentrations (10–100 μM) relevant to human disease (up to 150 μM in patients with CKD)[19]. TMAO-induced ERK1/2 activation was inhibited by a MEK inhibitor, suggesting the activation occurs downstream of Ras-Raf-MEK[50]. In addition, ERK1/2 activation in response to TMAO was suppressed when intracellular $Ca^{2+}$ was chelated (Supplementary Fig. 8B) or extracellular $Ca^{2+}$ was removed (Supplementary Fig. 8C). Moreover, $Ca^{2+}$ influx was sufficient to activate ERK1/2 in HRFs (Supplementary Fig. 8D), indicating that TMAO-induced $Ca^{2+}$ influx was required for ERK1/2 activation. We have reported that ERK1/2 phosphorylation was required for HRF trans-differentiation to myofibroblasts in response to TGF-β1[50]. Unlike short-term application (up to 30 min), TMAO stimulation for 48 h had minimal effect on ERK1/2 or SMAD3 phosphorylation (Fig. 5D and Supplementary Fig. 8E, F) or on the expression of the myofibroblast marker αSMA (Fig. 5E and Supplementary Fig. 8G). Conversely, TMAO dose-dependently augmented ERK1/2 pathway activation and myofibroblast trans-differentiation, when co-administered with TGF-β1 in comparison to TGF-β1 or TMAO alone, without affecting SMAD3 phosphorylation (Fig. 5D, E and Supplementary Fig. 8E, F).

### TMAO increases renal fibrosis: evidence from an intervention in male mice

To corroborate our in vitro findings suggesting that TMAO directly increases myofibroblast trans-differentiation and establish in vivo relevance, we performed Unilateral Ureter Obstruction (UUO) surgery in mice, a model assessing renal fibrosis in the absence of other co-morbidities[51] that could be also impacted by TMAO. Male mice were fed a choline (1% w/w)- or TMAO (0.12% w/w)-supplemented diet for 6 weeks[20]. Subsequently, the ureter of one kidney was ligated whilst the other kidney remained unobstructed (Supplementary Fig. 9A). In the unobstructed (control) kidneys, αSMA staining was not affected by diet. As expected, five days of UUO resulted in a significant increase in renal αSMA staining of the injured kidney. However, unlike the unobstructed kidneys, TMAO or choline diet supplementation resulted in significantly more myofibroblast expansion (Fig. 5F, G). Western blotting of kidney lysates corroborated augmented αSMA and vimentin (another marker of myofibroblasts[50]) expression in injured kidneys of mice receiving TMAO or choline diets (Fig. 5H–J). Similar to αSMA, collagen deposition or macrophage infiltration, another indicator of fibrotic kidney damage[46], were not affected by the TMAO or choline diets in the unobstructed kidneys (Supplementary Fig. 9B–E). Conversely, collagen and macrophage staining of kidney slides from UUO kidneys of mice consuming TMAO[3] or choline diets were significantly

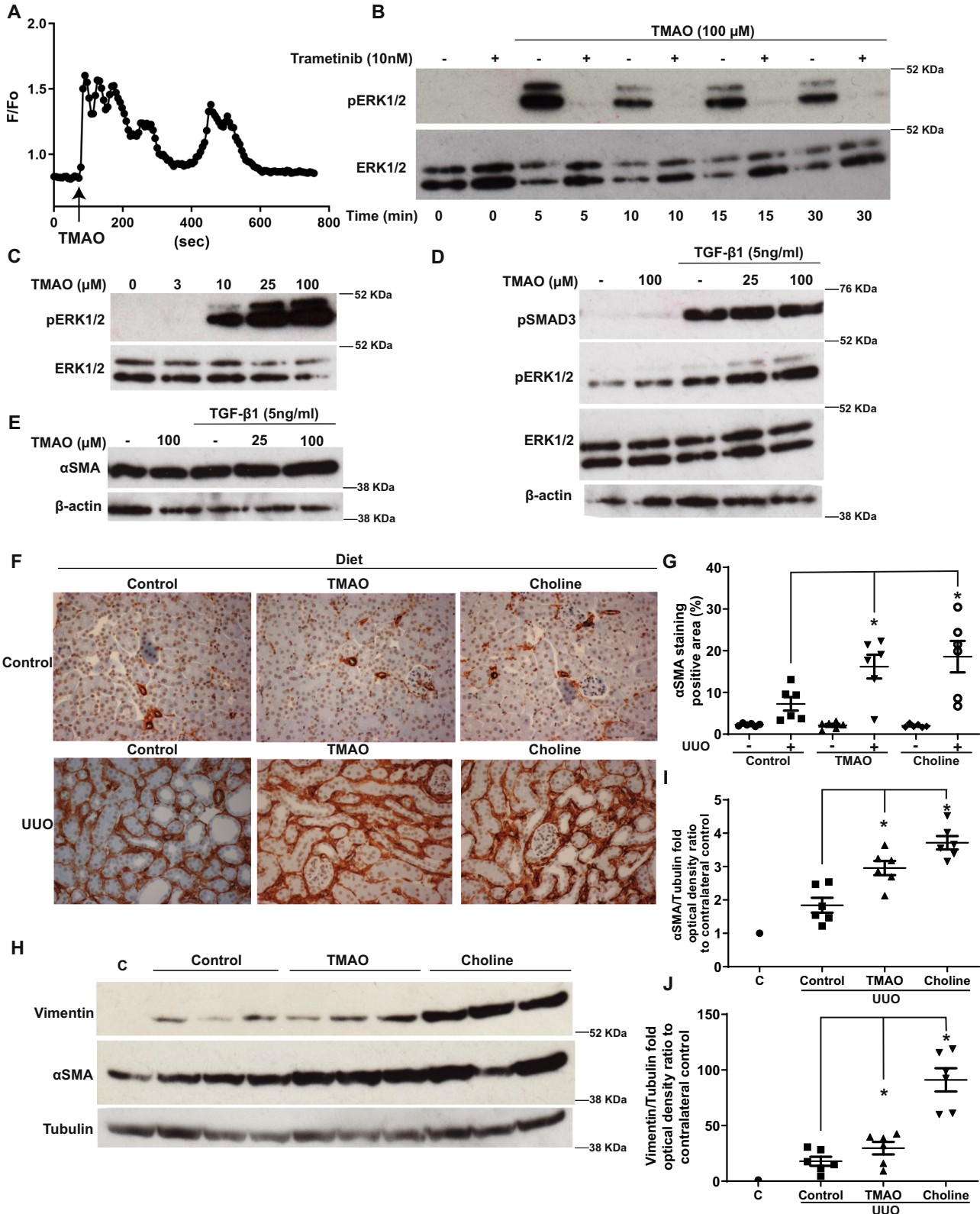

enhanced compared to controls. Thus, in our experiments in a murine model of kidney fibrosis, TMAO or choline diet supplementation resulted in "hyperactivation" of ERK1/2, mTORC1 and SMAD3 pro-fibrotic signaling[46,50] (Supplementary Fig. 9F).

Collectively, our in vivo and in vitro findings are consistent with TMAO aggravating kidney fibrosis due to ERK1/2 hyperactivation synergistically, probably with activation of the TGF-β1-mediated SMAD3 pathway causing a secondary hit in our disease model.

## Glucagon-like peptide-1 receptor analogs (GLP-1RAs) intake associates with lower serum TMAO concentration in MetaCardis T2D participants

Given the strong bidirectional connection we uncovered between serum TMAO and eGFR, we hypothesized that use of reno-protective medication could be linked with lower circulating TMAO. To identify suitable drugs, we trained algorithms to predict eGFR in MetaCardis patients with T2D, where TMAO appears to have the biggest impact on

**Fig. 5 | TMAO promotes myofibroblast differentiation and exacerbates renal fibrotic injury. A** Representative ratiometric traces (340/380 nm) from Human Renal Fibroblasts (HRFs) loaded with the Ca²⁺ indicator Fura-2 and stimulated with 100 μM TMAO. **B** Serum-starved HRFs were preincubated with the MEK inhibitor trametinib (10 nM) for 30 min prior to stimulation with 100 μM TMAO for the indicated times. Phospho-ERK1/2 levels were probed by Western blot; membranes were stripped and re-probed for total ERK1/2. **C** Serum-starved HRFs were stimulated with the indicated concentrations of TMAO and phospho-ERK1/2 and total ERK1/2 levels were determined as in (**B**). HRFs in complete medium were preincubated with the indicated concentrations of TMAO for 30 min and stimulated with TGF-β1 (5 nM) or vehicle for 24 h. Phospho-ERK1/2, phospho-SMAD3 (**D**) and alpha-smooth muscle actin (αSMA) (**E**) levels were probed with western blot. B-actin levels for (**D**, **E**) were probed in parallel western blots. For B–E a representative image from three independent biological repeats is shown. **F** Immunostaining with

αSMA of kidney sections (×20 magnification) from obstructed (UUO; 5 days post-surgery) or contralateral sham-operated (control) kidneys. Animals were fed normal chow (control), a diet containing 0.12% w/w TMAO (TMAO) or 1% choline w/w (Choline) for 6 weeks prior to surgery, as indicated. *n* = 6 per group. **G** Quantification of positive αSMA staining as (%) of stained area/field of view averaged from 5 images per animal. **H** Western blot of whole-kidney lysates for αSMA and vimentin expression. Tubulin, as loading control, was probed in parallel western blots. A representative photomicrograph from *n* = 2 Western blots with *n* = 1 animals for control (non-ligated kidneys, chow diet; **C**) *n* = 6 animals for all other groups is shown. OD of the (**I**) αSMA and (**J**) vimentin bands in (**H**) normalized against tubulin. The normalized density of the sham-control samples was arbitrarily set to 1. For all graphs, error bars represent the mean ± SEM of data from *n* = 4–6 animals per group. \**P* < 0.05 versus the UUO control. Source data are provided as a Source Data file.

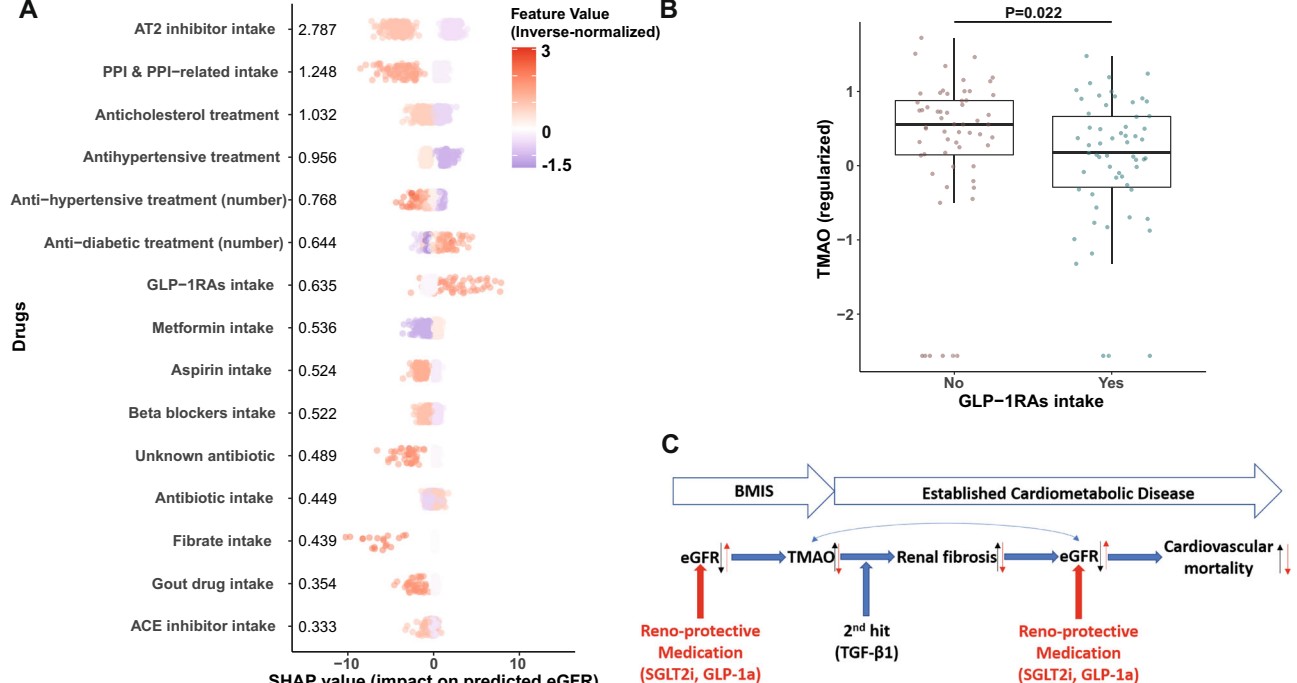

**Fig. 6 | Reno-protective medication is associated with reduced circulating TMAO in MetaCardis participants with T2D. A** Swarm plots of impact on model eGFR predictions (SHAP values) for MetaCardis T2D individuals (*N* = 561) for the top 15 drugs, as determined by xgboost algorithms trained exclusively on prescribed medication. Mean absolute SHAP values from participants with T2D are shown (in descending order) next to each variable. Individual dots, representing each participant, are colored by the inverse-normalized value of the corresponding drug variable. **B** Comparison of circulating regularized TMAO levels between subjects with T2D prescribed GLP-1 receptor agonists (GLP-1Ras; *N* = 59) and non-GLP-1Ras treated subjects with T2D propensity-matched for age, sex, disease group, and hypertension status (*N* = 59) (Supplementary Table 6 for group characteristics).

*P* value determined by two-sided Mann–Whitney *U* test. Center lines denote medians, box limits indicate the 25th and 75th percentiles, whiskers extend to the minimal and maximal values. **C** Summary of the main findings of our study. We demonstrate that eGFR, irrespective of disease stage, is the primary modifiable modulator of circulating TMAO. Far from being a bystander, TMAO significantly accelerates the rate of renal output decline by age, with its effect increasing at advanced stages of disease. TMAO promotes renal fibrosis in conjunction with established pathophysiology (two-hit" model) further negatively impacting renal clearance. Accordingly, medication with reno-protective properties (red arrows), such as GLP-1RAs, reduce circulating TMAO levels thereby potentially moderating its adverse effect on kidney function. Source data are provided as a Source Data file.

kidney function, using prescribed medication as input. SHAP analysis revealed that antihypertensive and anti-cholesterol treatments had a negative impact on predicted eGFR, probably reflecting more advanced disease (Fig. 6A). Conversely, anti-diabetic treatments had a positive effect, with GLP-1RAs being the drugs with the biggest positive impact on predicted eGFR (Fig. 6A). In accordance with their documented reno-protective effect even in T2D patients with no preexisting nephropathy[25], patients with T2D receiving GLP-1RAs had significantly lower circulating TMAO (Fig. 6B) than individuals matched for age, sex, hypertension, and disease group (Supplementary Fig. 8A–E and Supplementary Table 6). This exploratory analysis

suggests that GLP-1RAs may reduce serum TMAO concentration and thereby the associated higher risk of IHD.

## Discussion

With our approach combining epidemiological studies, explorative cellular experiments and murine interventions, we demonstrate that kidney function is the main modifiable factor consistently regulating fasting serum TMAO concentrations and that TMAO adversely impacts on eGFR, at least partially, by increasing kidney scarring synergistically with existing pathophysiology elevating kidney tissue TGF-β1 signaling. Consistent with our findings, use of reno-protective drugs,

GLP-1RAs[25], was associated with lower circulating TMAO in MetaCardis participants with T2D (Fig. 5C).

Irrespective of cardiometabolic disease severity, gut microbiota composition had a modest ($R^2 < 0.01$), albeit significant, association with serum TMAO levels in the MetaCardis population, in agreement with recent human studies[3,38]. We replicated associations of Firmicutes[3–5] with circulating TMAO and uncovered novel associations between an unknown bacterium (CAG01909) and *R. timonensis*, a marker of less diverse diet in autistic children[52]. However, taxa associated with TMAO diverged between disease modalities. This could at least, partially, be due to medication, suggesting that targeting TMAO production at the species level would be ineffective.

In agreement with a recent study[3], we did not find any significant association between habitual consumption of red meat and fasting serum levels of TMAO. With a few notable exceptions[3], the contribution of diet and in particular red meat and *L*-carnitine to TMAO levels, showing an increase following intervention, has predominately been examined in metabolically healthy volunteers[9–11]. In such interventions, often *L*-carnitine has been provided as a dietary supplement, which is poorly absorbed in the small intestine (-12%), as opposed to dietary *L*-carnitine (-71%)[53], and therefore may be more available for microbial catabolism in the upper gut and large intestine, leading to overestimating its role in TMA and, thereby, TMAO production. Our observations, similar to the report in ref. 3, suggest that in non-interventional settings where individuals habitually consume meat (75 to 233 g/day in European adults[37]), this contributes minimally to fasting circulating TMAO variability, possibly limiting the isolated effect of dietary manipulation on TMAO levels in non-interventional settings, aside strict vegans or vegetarians.

Further, our analyses revealed that the multi-omics signature associated with higher circulating TMAO concentrations and, therefore, elevated cardiovascular risk, shifted from obesity with increased insulin resistance toward overt T2D and IHD with only three variables (age, eGFR, and serum butyryl-carnitine) consistently being strong contributors across models built with the BMIS, T2D and IHD disease groups. This is in agreement with our recent report that the majority of markers of dysmetabolism manifest early during cardiometabolic disease development[31]. Accumulation of butyryl-carnitine has been associated with abnormal mitochondrial lipid β-oxidation in T2D[54], suggesting a link between TMAO and mitochondrial dysfunction consistent with the finding that TMAO binds to protein kinase R-like endoplasmic reticulum kinase and increases mitochondrial stress[55].

Far from being a bystander, mediation analysis indicated that TMAO significantly accelerated the rate of kidney function decline by age, with its effect increasing in more severe disease stages. Consistent with this, our preclinical work uncovered that TMAO primed renal fibroblasts for conversion to myofibroblasts, the primary collagen-producing cells in the kidney[46], and contributed to renal fibrosis, a hallmark of kidney damage irrespective of the underlying cause that significantly contributes to eGFR decline[46]. However, TMAO insult alone was not sufficient to convert renal fibroblasts to myofibroblasts requiring synergy with a preexisting pathological state, i.e., pro-fibrotic signaling ("second-hit") in the form of TGF-β1 stimulation, the most prominent pro-fibrotic cytokine[46]. TGF-β1 expression is increased in ligated UOO mouse kidneys[56] or kidneys of patients with diabetic nephropathy[57] and circulating TGF-β1 is elevated in conditions that are risk factors for CKD including hypertension, dyslipidemia and T2D[58]. Therefore, our results suggest that TMAO accelerates eGFR decline in concert with existing pro-fibrotic signaling (i.e., TGF-β1; "second-hit"), consistent with observations in animal models of CKD[44,45] and humans[20,21].

Supporting our assertion that TMAO accelerates cardiovascular disease progression together with existing pathology, at least partially by impacting the kidney, TMAO increased all-cause mortality only in individuals with eGFR <90 mL/min/1.73 m² (see ref. 59) and circulating

TMAO levels in healthy adults were not indicative of future atherosclerotic burden[60].

TMAO is an independent risk factor of cardiovascular morbidity and mortality in patients with established IHD or CKD[18,19] with accumulating evidence suggesting direct causal effects[47]. Our work suggests that reno-protective strategies could potentially lower circulating TMAO and therefore preserve kidney function in individuals with high circulating TMAO in the presence of risk factors known to increase TGF-β1 signaling (i.e., hypertension and T2D[58]). Indeed, we observed that patients with T2D prescribed GLP-1RAs, drugs with documented reno-protective and beneficial cardiovascular effects[24,25], had significantly lower fasting plasma TMAO levels than propensity-score matched controls. Differences in microbiota composition were predictive of glycemic responses to GLP-1RA intake[61] and further work is required to determine factors influencing GLP-1RA-mediated reno-protection which appear to be independent of improvements in glycemic control[62].

Collectively, our findings demonstrate that eGFR, irrespective of disease stage, is the primary modifiable modulator of circulating TMAO, which then by promoting renal fibrosis in conjunction with established pathophysiology ("second-hit" model) further negatively impacts kidney function. Accordingly, we observe that intake of GLP-1RAs is associated with lower circulating TMAO levels, thereby potentially moderating the adverse effect of TMAO on kidney function and suggesting a putative mechanism for these drugs that observed renoprotection in large pharmaceutical trials[25].

Our work conceptually advances the understanding of how TMAO levels (and therefore associated cardiovascular risk) are regulated in humans with a wide range of cardiometabolic disease burden in non-interventional settings. In addition, we uncover a direct mechanistic link between TMAO and renal fibrosis in conjunction with existing co-morbidities (known to elevate TGF-β1 signaling) such as hypertension and T2D. Furthermore, our findings suggest that therapeutic modalities preserving kidney function could markedly benefit and reduce cardiovascular risk in individuals with high circulating TMAO in the presence of risk factors (T2D, hypertension, or metabolic syndrome). This merits urgent testing in a longitudinal independent clinical trial.

## Methods
### MetaCardis study design and recruitment
MetaCardis is a cross-sectional study that recruited individuals at increasing stages of dysmetabolism and IHD severity (ranging from metabolically healthy, metabolic syndrome and/or obesity, T2D, IHD), aged 18–75-years old and recruited from Denmark, France and Germany between 2013 and 2015. Patients under care of the participating hospitals meeting the inclusion criteria of the study were invited to enroll. Healthy controls were recruited via public advertisement. Study participants provided written informed consent and the study was undertaken according to Helsinki Declaration-II. Ethical approval was obtained from the Ethics Committee CPP Ile-de France, the Ethical Committees of the Capital Region of Denmark (H-3-2013-145), and Ethics Committee at the Medical Faculty at the University of Leipzig (047-13-28012013). Study design, recruitment and exclusion criteria has been extensively described[26,31,41,63]. The overarching goal of the trial was to investigate the impact of qualitative and quantitative changes in the gut microbiota on the pathogenesis of cardiometabolic diseases (CMDs) and their associated co-morbidities (ClinicalTrials.gov Identifier: NCT02059538). For the present study, patients were subclassified in three groups: BMI-spectrum patients (BMIS[26]; $N = 837$), encompassing MetaCardis participants presenting with metabolic syndrome-related risk factors or conditions (hypertension, as defined by the American Heart Association[64]; obesity, as defined by the World Health Organization[65] and metabolic syndrome, as defined by the International Diabetes Federation[66]) and patients diagnosed with type-2 diabetes (T2D, as defined by the American Diabetes Association[67]; $N = 561$)

or ischemic heart disease (IHD; $N = 356$). The IHD group comprised patients with Acute (<15 days) Coronary Syndrome (ACS; $N = 106$), Chronic IHD (CIHD; $N = 157$) with normal Left Ventricular Ejection Fraction (LVEF) determined by echocardiography and Heart Failure patients (HF; $N = 93$, LVEF < 45%). Cardiometabolic disease is used as an umbrella term for all the above cases, and severity of cardiometabolic disease refers, in this manuscript, to the progression from single risk factors such as obesity to overt T2D and cardiac phenotype (ischemic heart disease and heart failure).

## Sample and phenotypic information collection

Biofluid and biomatter collection has been described elsewhere[26,31,41,63]. Briefly, blood samples were collected in the morning after overnight fasting, and fecal samples were collected at home by participants, frozen immediately and transferred to study centers on dry ice within 48 h. All samples were stored at −80 °C until use. Clinical history, medication and phenotypic information were acquired as described[26,31,41,63] with standardized procedures across centers. Participants reported habitual food intake through a customized Food Frequency Questionnaire (FQQ). The participant's responses to the FQQs was validated by three web-based dietary recalls for a subset of study participants ($N = 324$) according to established practices[35]. Bioclinical variables were measured in a single center according to standard procedures[31]. Estimated glomerular filtration rate (eGFR) was calculated with the CKD-EPI formula without ethnicity adjustment[68].

## Metabolic profiling

**¹H-nuclear magnetic resonance (¹H-NMR) spectroscopy.** Spectra acquisition, using an Avance spectrometer (Bruker) at 600 MHz; and structural assignments have been extensively described previously[31]. Absolute quantifications were derived using the "In Vitro Diagnostics for research" (IVDr) quantification BI-Quant-UR algorithm (Bruker, https://www.bruker.com/en/products-and-solutions/mr/nmr-clinical-research-solutions/b-i-quant-ur.html, v1.1). For some analyses metabolites absolute quantifications were divided by the creatinine concentration, also derived by the IVDr platform from the ¹H-NMR spectra.

**Gas chromatography-coupled mass spectrometry (GC-MS).** Serum samples (100 µl) were prepared, analyzed and processed as described[31,41]. Briefly, protein was methanol-precipitated, methanol was evaporated to dryness and subsequent to derivatization samples were injected to an Agilent 7890B-5977B Inert Plus GC-MS system. The chromatographic column was an Agilent ZORBAX DB5- MS (30 m × 250 µm × 0.25 µm + 10 m Duragard). The temperature gradient was 37.5 min long and the mass analyzer was operated in full-scan mode between 50 and 600 m/z. Peaks were annotated with the use of the Fiehn library (Agilent G1676AA Fiehn GC/MS Metabolomics RTL Library, User Guide, Agilent Technologies, https://www.agilent.com/cs/library/usermanuals/Public/G1676-90001_Fiehn.pdf). Metabolic features with low reproducibility or linearity were removed from the dataset, resulting in 102 annotated metabolic features.

**Ultra-performance liquid chromatography-tandem mass spectrometry (UPLC−MS/MS).** UPLC−MS/MS performed on a Waters Acquity UPLC-Xevo TQ-S UPLC−MS/MS system equipped with an Acquity BEH HILIC (2.1 × 100 mm, 1.7 µm) chromatographic column was employed to determine TMA, TMAO, choline, betaine, γ-butyrobetaine, betaine aldehyde, butyryl-carnitine, isovaleryl-carnitine, OH-isovaleryl-carnitine, stearoyl-carnitine, oleoyl-carnitine, linoleoyl-carnitine, myristoyl-carnitine, lauroyl-carnitine, and decanoyl-carnitine as described previously[31,41]. TMAO circulating values of all MetaCardis participants were log₁₀-transformed, and subsequently the median was subtracted and divided by the standard deviation[23] (SD; regularized TMAO values throughout). MassLynx™ software (Waters corporation, v4.2) was used for data acquisition and analysis.

## Metagenomic analysis

Phylogenetic microbiota profiles were built after correction for bacterial load as extensively described[26,31,41,63] using a protocol devised in ref. 69 with modifications. Briefly, total fecal DNA was extracted following the International Human Microbiome Standards (IHMS) guidelines (SOP 07 V2 H) and samples were sequenced using ion-proton technology (ThermoFisher Scientific). Gene abundance profiling was performed using the 9.9 million gene integrated reference catalog of the human microbiome, as described[26,31,41,63]. The following libraries were used to process metagenomic data: METEOR (v.3.2; https://forgemia.inra.fr/metagenopolis/meteor), Alientrimmer v0.4.0, Bowtie2 v2.3.4, MetaOMineR (momr, v1.31), Omixer-RPM (v1.0).

## Customized microbial module analysis (GMM)

Manually curated customized module sets focusing on anaerobic bacterial and archaeal fermentation processes relevant to the human gut microbiota were assembled as previously extensively described[26,31,41,63].

## Statistical analyses

Statistical analysis was undertaken using R (v4.03; R Core Team (2020). R: A language and environment for statistical computing. R (v4.03) Foundation for Statistical Computing, Vienna, Austria. URL: https://www.R-project.org/). For comparing two groups, we used Mann–Whitney $U$ and for multiple group comparisons Kruskal–Wallis tests. Unadjusted Spearman correlations were computed using R, whilst adjusted Spearman correlations with ppcor (v1.1). $P$ values were corrected for multiple comparisons using the Benjamini–Hochberg method.

## Machine learning (ML) analysis

**Variable groups.** Patient phenotypic variables were separated into ten groups (Supplementary Data 1). Specifically, biological parameters included biochemical and clinical serum laboratory tests, including lipids, glycated hemoglobin, creatinine, and eGFR. Clinical parameters consisted of clinical history, BMI, systolic and diastolic blood pressure, anthropometric variables, and stool frequency and type. Demographic information comprised age, physical activity, educational and income levels, smoking status, ethnicity and country of recruitment. Drug variables included intake of common medication as described[41], number of antibiotic courses in the last 5 years and number of anti-hypertensive, anti-diabetic and lipid-lowering treatments. Dietary parameters included habitual consumption of 37 food items, daily nutrients intake derived from these food items calculated as in[35], alternative Healthy Eating[70] (aHEI), Dietary Approaches to Stop Hypertension[71] (DASH) and Dietary Diversity[72] (DDS) scores. Clinical risk factors variables included age, systolic and diastolic blood pressure, glycated hemoglobin levels, fasting cholesterol levels, smoking status and waist circumference as described[31]. Serum metabolomics comprised the absolute or relative levels of 116 circulating metabolites determined by GC-MS or UPLC−MS/MS[31,41]. Urine metabolomics included absolute quantification of 47 urine metabolites from ¹H-NMR spectra calculated with the IVDr algorithm. Microbiota variables included an abundance of 699 bacterial species present in at least 20% of MetaCardis patients corrected for microbial load and the first 10 principal components from a PCA of relative microbial gene abundances[31,41]. Microbiome (Modules) group included abundance of 116 manually curated bacterial modules[31]. In all cases, categorical variables were converted into dummy variables using caret (v6.0.86).

**Boosted decision trees (Xgboost).** We predicted regularized circulating TMAO levels by using gradient-boosting decision trees based on the xgboost algorithm (v1.3.2.1)[73], co-opting a strategy from ref. 23. Xgboost consistently outperforms other algorithms in Kaggle competitions for tabular data. For each of our 10 variable groups

(Supplementary Table 5) we optimized xgboost models using fivefold cross-validation and two sequential hyperparameter grids searches with early stopping (972 different parameter combinations for each feature group in total) to predict mean-centered and unit variance-scaled (regularized) TMAO levels in the left-out group using root-mean-square error (RMSE) to evaluate model outcomes. After parameter optimization, we predicted circulating TMAO using 5-fold cross-validation for 100 iterations using as input the variables of each group or all variables (full model). For each round, we calculated the coefficient of determination using the rsq function from yardstick (v0.0.7) and the predicted regularized TMAO values. Xgboost models trained with 80% of each patient group participants during cross-validation (five for each round) were saved and used for feature attribution analysis in the left-out group (see below).

To minimize the risk of overfitting, we took four steps:

1. during the model parameter optimization step, training was stopped if predictions were not improved for ten rounds (early_stopping_rounds = 10).
2. We incorporated the regularization parameters lambda and gamma in our models, thus making our models more conservative[23].
3. We introduced randomness by using 0.8–0.9 of the available variables in each variable group (colsample_bytree=0.8–0.9 depending on parameter optimization; Supplemental Tables 4–6 for specific model parameters) for the training of each tree, as a way to minimize overfitting.
4. all our conclusions are based on ensemble models (the average of 100 independent runs) that in combination with the introduced randomness and the out-of-sample predictions (fivefold cross-validation) makes our models conservative.

Please see also the XGBoost documentation for additional information on model parameters (https://xgboost.readthedocs.io/en/latest/index.html).

**SHapley additive exPlanations (SHAP) analysis.** We interpreted our ML models and assigned relative importance to variables influencing circulating TMAO levels by co-opting SHAP values, as expanded for tree-based ML models[28] and recently used to objectively evaluate factors driving metabolite plasma levels in humans[23]. Briefly, for each prediction the SHAP value of a particular variable is the difference in the model output when this variable is included versus when excluded. Variables are added to the model in all possible orderings, and the SHAP value is computed from the average of model outcomes[28]. Variable SHAP values for each individual in the five left-out groups were extracted from corresponding xgboost models trained on the remaining 80% participants for each variable group using SHAP-forxgboost (0.1.0) and averaged over 100 iterations. We assigned relative importance to each variable by computing the mean of the absolute SHAP value for all individuals in each disease group similarly to ref. 23. For swarm plots depicting each individual participants SHAP values, variable values were inverse-normalized using the RNOmni (v1.0.0) package.

**Least absolute shrinkage and selection operator (LASSO).** We used the LASSO algorithm as implemented in the glmnet (v4.1) package to predict regularized TMAO values in BMIS ($N = 582$) participants using all available variables as input (full model). Initially, we used fivefold cross-validation to determine the best regularization parameter lambda (from 100 values ranging from $10^{-3}$ to $10^5$) for predicting circulating TMAO in the left-out group. We next trained five LASSO regression models in 80% of BMIS participants excluding the corresponding left-out group with the optimal lambda parameter. These models were then used to predict regularized TMAO in the left-out groups and subsequently to calculate the coefficient of determination

using the rsq function from yardstick (v0.0.7). This process was repeated for 100 iterations, similar to the xgboost analysis.

**Clustering.** Clustering was performed using the built-in R k-means function with the Hartigan−Wong algorithm using 25 random sets, as described[27].

**Multivariate analysis.** To identify microbiota composition differences between individuals split in TMAO clusters we performed multivariate homogeneity analysis of Bray−Curtis dissimilarity matrices at the species level (699 species present in at least 20% of our population) and determined statistical significance with a permutation ANOVA test (999 iterations) using the vegan (v2.5.7) package. Permutation analysis of variance (PERMANOVA) of regularized circulating TMAO versus Bray−Curtis taxonomic dissimilarity matrices with age, sex, and country of recruitment as covariates for 999 iterations were performed with vegan (v2.5.7).

**Mediation analysis.** We performed mediation analysis[74] to assess the putative impact of eGFR on plasma TMAO increase with age and, conversely, the impact of TMAO on eGFR decline with age. We used the Preacher and Hayes bootstrapping method, as implemented in the mediation (v4.5.0) R package[75], using general linear models with sex and country of recruitment as covariates. Confidence intervals and Bayesian *P* values were computed after 999 simulations.

**Propensity-score matching.** MetaCardis participants with T2D were propensity-score matched using the R package MatchIt[76] (v.4.1.0) with age, sex, disease group, and hypertension status as covariates using nearest neighbor matching determined by generalized linear models. All covariates were given equal weight.

### Preclinical models
#### In vitro experiments
**Cell culture and protein extraction.** Primary human adult kidney fibroblasts from a single donor (DV Biologics, AU009-F) were cultured in 10% FCS low-Glucose DMEM with 1% Pen/Strep antibiotics (Sigma). For acute (up to 30 min) TMAO stimulation, cells (100,000/condition) were serum-starved for 1 h in a physiologic serum-free buffer as previously described[50]. Fibroblasts were preincubated with Tramenitib (10 nM, Shelleckchem), BAPTA-AM (20 μM, Molecular Probes) or vehicle for 30 min before TMAO (Sigma) stimulation. For longer-term (24 h) stimulation, fibroblasts were in serum-free low-glucose DMEM overnight and subsequently stimulated with TMAO, TGF-β1 (5 ng/ml, R&D), or their combination. At the end of the experiment, fibroblasts were lysed and stored at −80 °C until further use, as described[50].

**Cytosolic [Ca²⁺] measurements.** Fibroblasts were serum-starved overnight in low-glucose DMEM and subsequently loaded with 5 μM fura-2-AM (Molecular Probes) in pH 7.4 Hanks Balanced Salt Solution (HBSS; Sigma) containing $Ca^{2+}$ and $Mg^{2+}$. Measurements were obtained on an epifluorescence inverted microscope equipped with a ×20 fluorite objective. Single-cell intracellular $Ca^{2+}$ ($[Ca^{2+}]_i$) was monitored using excitation at 340 and 380 nm, through a monochromator (Cairn Research). Emitted light was reflected through a 515 nm filter to a QImaging Retiga CCD camera (Cairn Research) and digitized to 12-bit resolution. All imaging data were collected and analyzed using software from Andor.

**Animal procedures.** All animal experiments were conducted in accordance with the United Kingdom Home Office Animals 1986 Scientific Procedures, with local ethical committee approval (Project License 70/8356). Male C57BL/6 J mice (Charles River) at 6–8 weeks of age were fed a control, 0.12% w/w TMAO- or 1% w/w choline-containing diets (Teklad Global 18% Protein Rodent Diet (Cat. #2018); 6 animals/

group) for 6 weeks[20]. Food and water were available ad libitum, and mice were held at 12/12 h light/dark cycle at 20–23 °C and 40–60% humidity in standard individually ventilated cages (5–6 animals in each cage). Subsequently, Unilateral Ureter Obstruction (UUO) was established as previously described[50]. Only male mice were used, since this is common practice for this model[50]. Briefly, mice were anesthetized with isoflurane and the abdominal cavity was exposed using midline laparotomy, the right ureter was isolated and tied off 0.5 cm from the pelvis. The left ureter was left unclamped and served as the sham-operated control. At day 5, mice were euthanized by terminal anesthesia with ketamine and cardiac puncture, and subsequently the kidneys were harvested. Both kidneys from each animal (UUO and sham) were cut in half longitudinally. One-half of each kidney was snap-frozen in liquid nitrogen and subsequently stored at −80 °C for Western blotting. The other half was fixed with 10% Formalin (Sigma) for 16 h at 4 °C, transferred to a 70% v/v ethanol solution for a further 24 h and was then paraffin-embedded for immunohistochemistry.

**Western blot.** We used the NuPAGE electrophoresis and buffer system (Invitrogen) for immunoblot analysis of kidney samples or cellular lysates as described[48,49]. Amersham™ Rainbow™ full-range molecular weight (MW) markers (Merck, GERPN800E) were used to determine MW and cut membranes for probing with specific antibodies. Proteins were visualized with ECL Prime (GE Healthcare). Optical densities of bands of interest were determined using ImageJ 1.46r (NIH) and normalized against loading controls. The value of the normalized control sample was arbitrarily set to 1. Membranes were stripped using the Restore Plus reagent (Fisher Scientific), and re-probed with appropriate loading controls (total ERK1/2). In cases where there was overlap in MW between loading controls and proteins of interest, loading controls were run in parallel gels. The following antibodies were used in the present study with dilutions in parentheses: From Cell Signaling Technology, Phospho-p44/42 MAPK (Erk1/2) (Thr202/Tyr204) Antibody (1:3000) #9101, p44/42 MAPK (Erk1//2) Antibody (1:5000) #9102, Phospho-Smad3 (Ser423/425) (C25A9) Rabbit mAb (1:1000) #9520, Phospho-p70 S6 Kinase (Thr389) Antibody (1:1000) #9205, Phospho-S6 Ribosomal Protein (Ser235/236) Antibody (1:1000) #2211, Phospho-4E-BP1 (Ser65) Antibody (1:1000) #9451, Vimentin (R28) antibody (1:1000) #3932. From Sigma-Aldrich, monoclonal anti-actin, α-Smooth Muscle (clone 1A4) A2547 (1:10,000), monoclonal Anti-α-Tubulin antibody (clone DM1A; 1:10,000) T9026, anti-β-actin antibody (clone AC-74, 1:10,000) A2228.

**Immunohistochemical staining.** Formalin-fixed kidneys were embedded in paraffin, and 4-μm sections were cut by the Barts Cancer Institute Pathology Unit. Staining for Sirius red, α-Smooth Muscle Actin (αSMA), and F4/F80 were performed as described[50] with the DISCOVERY XT (Ventana) automated slide processing instrument using the OmniMap reagents (Ventana), according to the manufacturer's recommendations. Images were captured at ×20 magnification, using a Zeiss AxioPhot microscope with an AxioCam HRc camera. Five kidney cortex images were captured per mouse, and staining was quantified as the percentage of the total area, using ImageJ 1.46r (NIH). The Anti-F4/F80 (Cl:A3-1) antibody (1:50) from Biorad (MCA497) was used.

**Reporting summary**
Further information on research design is available in the Nature Portfolio Reporting Summary linked to this article.

## Data availability
The Raw shotgun sequencing data that support the findings of this study have been deposited in the European Nucleotide Archive under accession codes "PRJEB37249", "PRJEB38742", "PRJEB41311", and "PRJEB46098". The Serum NMR and urine NMR metabolome data have been uploaded to Metabolights with accession number "MTBLS3429".

The Serum GC-MS and isotopically quantified serum metabolites (UPLC–MS/MS) that have been used in this study are deposited in MassIVE with accession numbers "MSV000088042 [https://doi.org/10.25345/C5CV76]" and "MSV000088043 [https://doi.org/10.25345/C58246]", respectively. In adherence to EU and national privacy laws, unrestricted access to patient phenotypic data cannot be provided for this study. Interested researchers, wishing to access individual phenotypic data would need to submit argued applications to the relevant National Data Protection Agencies. These are the Danish Data Protection Agency (https://www.datatilsynet.dk/english) for phenotypic data from study participants recruited in Denmark, the Federal Commissioner for Data Protection (https://www.bfdi.bund.de/EN/Home/home_node.html) for phenotypic data from study participants recruited in Germany and the Commission Nationale Informatique & Libertés (https://www.cnil.fr/en/home) for phenotypic data of study participants recruited in France. Application procedures are given on the outlined websites. If such permission is granted, phenotypic data will be then made available by the corresponding authors within 5 weeks. Source data are provided with this paper.

## Code availability
No custom code or algorithm was used for the analyses conducted in this work. Code and associated phenotypic data to replicate the analyses presented in this work can be obtained by contacting the corresponding authors (please also see the Data availability statement).

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

## Acknowledgements

This work was supported by European Union's Seventh Framework Program for research, technological development, and demonstration under grant agreement HEALTH-F4-2012-305312 (METACARDIS). Assistance Publique-Hôpitaux de Paris (AP-HP) is the promoter of clinical investigation (MetaCardis). Partial funding supports to K.C. and J.A.-W. were also obtained from Leducq Foundation (TransAtlantic grant), SFN (Société Française de Nutrition), F-CRIN-FORCE network for support, INSERM via ITMO and JPI-Microdiet study and Novo Nordisk foundation (Jacobaus prize). S.K.F. received support from Deutsche Forschungsgesellschaft SFB1365 ("RENOPROTECTION") and SFB1470: "HFpEF". P.A. and K.Ch. are recipients of a Wellcome ISSF Fellowship (ISSF204834/Z/16/Z). M.-E.D. is supported by the NIHR Imperial Biomedical Research Centre, GutsUK, Diabetes UK and by grants from the French National Research Agency (ANR-10-LABX-46, European Genomics Institute for Diabetes), from the National Center for Diabetes Precision Medicine—PreciDIAB, which is jointly supported by the French National Agency for Research (ANR-18-IBHU-0001), by the European Union (FEDER), by the Hauts-de-France Regional Council (Agreement 20001891/NP0025517) and by the European Metropolis of Lille (MEL, Agreement 2019_ESR_11) and by Isite ULNE (R-002-20-TALENT-DUMAS), also jointly funded by ANR (ANR16-IDEX-0004-ULNE), the Hauts-de-France Regional Council (20002845) and by the European Metropolis of Lille (MEL). This research was conducted within the context of the CNRS–Imperial International Research Project METABO-LIC. The Novo Nordisk Foundation Center for Basic Metabolic Research is an independent research institution at the University of Copenhagen partially funded by an unrestricted donation from the Novo Nordisk Foundation. We thank the subjects for their participation in the MetaCardis study and particularly patient associations (Alliance du Coeur and CNAO) for their input and interface, as well as Dr Dominique Bonnefont-Rousselot (Department of Metabolic Biochemistry, Pitié-Salpêtrière hospital) for the analysis of plasma lipid profiles. We thank the nurses, technicians, clinical research assistants and data managers from the Clinical investigation platform at the Institute of Cardiometabolism and Nutrition for patient investigations, at the CRNH (Centre de recherche en Nutrition Humaine CRNH-Ile de France) and, the Clinical Investigation Center (CIC) from Pitié-Salpêtrière Hospital for investigation of healthy controls. Quanta Medical provided regulatory oversight of the clinical study and contributed to the processing and management of electronic data.

## Author contributions

P.A., K.C., and M.E.-D. developed the present project concept and protocol. K.C. (coordinator), O.P., M.S., S.D.E., P.B., J.R., M.-E.D., F.B., and J.N. conceived the overall objectives and study design of the MetaCardis initiative. MetaCardis cohort recruitment, phenotyping and lifestyle: J.A.-W., R.C., T.N., J.-E.S., F.A., L.K., H.V., T.H., G.H., R.I., J.-M.O., and M.B. and supervised by K.C., M.S., and O.P. Data curation: R.C., S.K.F., J.A.-W., and T.N. Quantitative microbiota profiling: S.V.S. and G.F. Serum and urine metabolome profiling: A.M., J.C., M.T.O., and L.H. Biochemical analyses: J.P.G. and C.R. Bioinformatics and statistical analyses: P.A., S.K.F., G.F, S.V.S., R.A., E.L.C., L.P.C., E.P., E.B., D.G., K.Ch., P.F., F.P.-C., M.C., and J.-D.Z. Dietary FFQ analyses: S.A. and B.H. Modeling of phenotypic data: P.A. Cellular in vitro experiments: P.A. Murine in vivo experiments P.A. and J.K. with supervision from M.M.Y. The manuscript was written by P.A., K.C., and M-E.D. with input from J.-A.W., R.C., S.K.F., M.M.Y., M.S., O.P., and S.D.E. All authors participated in project development, discussion of results and revision of the article and approved the final version for submission.

## Competing interests

K.C. is a consultant for Danone Research, Ysopia, and CONFO therapeutics for work not associated with this study. K.C. held a collaborative research contract with Danone Research in the context of MetaCardis project. F.B. is a shareholder of Implexion Pharma AB. M.B. received lecture and/or consultancy fees from AstraZeneca, Boehringer-Ingelheim, Lilly, Novo Nordisk, Novartis, and Sanofi. The remaining authors declare no competing interests.

## Additional information

Petros Andrikopoulos [1,2,39] ✉, Judith Aron-Wisnewsky [3,4,39], Rima Chakaroun [5,6,39], Antonis Myridakis [1,7,39], Sofia K. Forslund [8,9,10,11,12,39], Trine Nielsen [13], Solia Adriouch [3], Bridget Holmes [14], Julien Chilloux [1], Sara Vieira-Silva [15,16,17,18], Gwen Falony [15,16,17,18], Joe-Elie Salem [4], Fabrizio Andreelli [3,4], Eugeni Belda [3,19], Julius Kieswich [20,21], Kanta Chechi [1,2], Francesc Puig-Castellvi [22], Mickael Chevalier [22], Emmanuelle Le Chatelier [23], Michael T. Olanipekun [1], Lesley Hoyles [24], Renato Alves [8], Gerard Helft [4,25], Richard Isnard [3,4], Lars Køber [26], Luis Pedro Coelho [8], Christine Rouault [3], Dominique Gauguier [27,28], Jens Peter Gøtze [29], Edi Prifti [3,19], Philippe Froguel [22,30], The MetaCardis Consortium*, Jean-Daniel Zucker [3,19], Fredrik Bäckhed [6,13], Henrik Vestergaard [13,31], Torben Hansen [13], Jean-Michel Oppert [4], Matthias Blüher [5], Jens Nielsen [32], Jeroen Raes [15,16], Peer Bork [8,10,33,34], Muhammad M. Yaqoob [20,21,40], Michael Stumvoll [5,40], Oluf Pedersen [13,35,40], S. Dusko Ehrlich [36,40], Karine Clément [3,4,40] ✉ & Marc-Emmanuel Dumas [1,2,22,28,40] ✉

[1]Section of Biomolecular Medicine, Department of Metabolism, Digestion and Reproduction, Imperial College London, London, UK. [2]Section of Genomic & Environmental Medicine, National Heart & Lung Institute, Imperial College London, London, UK. [3]Sorbonne Université, INSERM, Nutrition and obesities; systemic approaches (NutriOmics), Paris, France. [4]Assistance Publique Hôpitaux de Paris, Pitie-Salpêtrière Hospital, Paris, France. [5]Medical Department III—Endocrinology, Nephrology, Rheumatology, University of Leipzig Medical Center, Leipzig, Germany. [6]The Wallenberg Laboratory, Department of Molecular and Clinical Medicine, Institute of Medicine, Sahlgrenska Academy, University of Gothenburg, Gothenburg, Sweden. [7]Environmental Research Group, MRC Centre for Environment and Health, School of Public Health, Imperial College London, 86 Wood Lane, London W12 0BZ, UK. [8]Structural and Computational Biology, European Molecular Biology Laboratory, Heidelberg, Germany. [9]Experimental and Clinical Research Center, a cooperation of Charité-Universitätsmedizin and the Max-Delbrück Center, Berlin, Germany. [10]Max Delbrück Center for Molecular Medicine (MDC), Berlin, Germany. [11]Charité University Hospital, Berlin, Germany. [12]DZHK (German Centre for Cardiovascular Research), Partner Site Berlin, Berlin, Germany. [13]Novo Nordisk Foundation Center for Basic Metabolic Research, Faculty of Health and Medical Sciences, University of Copenhagen, Copenhagen, Denmark. [14]Danone Research, Palaiseau, France. [15]Laboratory of Molecular Bacteriology, Department of Microbiology and Immunology, Rega Institute, KU Leuven, Leuven, Belgium. [16]Center for Microbiology, VIB, Leuven, Belgium. [17]Institute of Medical Microbiology and Hygiene and Research Center for Immunotherapy (FZI), University Medical Center of the Johannes Gutenberg-University Mainz, Mainz, Germany. [18]Institute of Molecular Biology (IMB), Mainz, Germany. [19]Sorbonne Université, IRD, Unité de Modélisation Mathématique et Informatique des Systèmes Complexes, UMMISCO, F-93143 Bondy, France. [20]Diabetic Kidney Disease Centre, Renal Unit, Barts Health National Health Service Trust, The Royal London Hospital, London, UK. [21]Centre for Translational Medicine and Therapeutics, William Harvey Research Institute, Barts and the London School of Medicine and Dentistry, Queen Mary University of London, London, UK. [22]European Genomics Institute for Diabetes, EGENODIA, INSERM U1283, CNRS UMR8199, Institut Pasteur de Lille, Lille University Hospital, University of Lille, Lille, France. [23]Metagenopolis, INRA, AgroParisTech, Université Paris-Saclay, Paris, France. [24]Department of Biosciences, School of Science and Technology, Nottingham Trent University, Nottingham, UK. [25]Institute of Cardiometabolism and Nutrition, ICAN, INSERM, 1166 Paris, France. [26]Department of Cardiology, Rigshospitalet, University of Copenhagen, Copenhagen, Denmark. [27]INSERM UMR 1124, Université de Paris, 45 rue des Saint-Pères, 75006 Paris, France. [28]McGill Genome Centre, McGill University, 740 Doctor Penfield Avenue, Montreal, QC H3A 0G1, Canada. [29]Department of Clinical Biochemistry, Rigshospitalet, University of Copenhagen, Copenhagen, Denmark. [30]Section of Genetics and Genomics, Department of Metabolism, Digestion and Reproduction, Imperial College London, London W12 0NN, UK. [31]Department of Medicine, Bornholms Hospital, Rønne, Denmark. [32]Department of Biology and Biological Engineering, Chalmers University of Technology, Gothenburg, Sweden. [33]Department of Bioinformatics, Biocenter, University of Würzburg, Würzburg, Germany. [34]Yonsei Frontier Lab (YFL), Yonsei University, Seoul 03722, South Korea. [35]Center for Clinical Metabolic Research, Gentofte University Hospital, Copenhagen, Denmark. [36]Department of Clinical and Movement Neurosciences, University College London, London NW3 2PF, UK. [39]These authors contributed equally: Petros Andrikopoulos, Judith Aron-Wisnewsky, Rima Chakaroun, Antonis Myridakis, Sofia K. Forslund. [40]These authors jointly supervised this work: Muhammad M. Yaqoob, Michael Stumvoll, Oluf Pedersen, S. Dusko Ehrlich, Karine Clément, Marc-Emmanuel Dumas. *A list of authors and their affiliations appears at the end of the paper. ✉e-mail: p.andrikopoulos04@imperial.ac.uk; karine.clement@inserm.fr; m.dumas@imperial.ac.uk

# The MetaCardis Consortium

Rohia Alili[3], Ehm Astrid Andersson Galijatovic[13], Olivier Barthelemy[4], Jean-Philippe Bastard[3], Jean-Paul Batisse[4], Pierre Bel-Lassen[3], Magalie Berland[23], Randa Bittar[4], Hervé Blottière[23], Frederic Bosquet[4], Rachid Boubrit[4], Olivier Bourron[4], Mickael Camus[3], Cecile Ciangura[4], Jean-Philippe Collet[4], Arne Dietrich[5], Morad Djebbar[4], Angélique Doré[23], Line Engelbrechtsen[13], Leopold Fezeu[37], Sebastien Fromentin[23], Nicolas Pons[23], Marianne Graine[3], Caroline Grünemann[5], Agnes Hartemann[4], Bolette Hartmann[13], Malene Hornbak[13], Sophie Jaqueminet[4], Niklas Rye Jørgensen[29], Hanna Julienne[23], Johanne Justesen[13], Judith Kammer[5], Nikolaj Karup[13], Ruby Kozlowski[1], Michael Kuhn[8], Véronique Lejard[23], Ivica Letunic[8], Florence Levenez[23], Lajos Marko[10], Laura Martinez-Gili[1], Robin Massey[23], Nicolas Maziers[23], Lucas Moitinho-Silva[8], Gilles Montalescot[4], Ana Luisa Neves[1], Laetitia Pasero Le Pavin[23], Francoise Pousset[4], Andrea Rodriguez-Martinez[1], Sebastien Schmidt[8], Tatjana Schütz[5], Lucas Silva[8], Johanne Silvain[4], Mathilde Svendstrup[13], Timothy D. Swartz[3], Thierry Vanduyvenboden[23], Eric O. Verger[38] & Stefanie Walther[5]

[37]Nutritional Epidemiology Research Team (EREN), Centre of Research in Epidemiology and Statistics Sorbonne Paris Cité, Inserm (U1153), Inra (U1125), Cnam, Paris 13 University, COMUE Sorbonne Paris Cité, 93017 Bobigny, France. [38]MoISA, Univ Montpellier, CIHEAM-IAMM, CIRAD, INRAE, Institut Agro, IRD, Montpellier, France.

