## [Peer Review File · Nature Communications]

Evidence of a causal and modifiable relationship between kidney function and circulating trimethylamine N-oxideREVIEWER COMMENTS

Reviewer #1 (Remarks to the Author):

In this manuscript, authors recruited 837 BMI-spectrum (BMIS) patients, 561 type II diabetes (T2D) patients and 337 patients with ischemic heart disease, and collected diet information and tissues to measure serum metabolites, urine metabolites and do metagenomics sequencing. Authors used “explainable” machine learning and tested each monitored parameter in predicting fasting plasma TMAO variance and found that age is the largest factor, then kidney function and its related parameters, and diet and microbiome’s contribution are very small. Authors also compared serum TMAO in T2D with prescribed new-generation anti-diabetics (GLP-1 Receptor Agonists; GLP-1RAs) with evidenced reno-protective effects vs without anti-diabetics in the collected cohort and found that the former group had significantly lower serum TMAO than the latter group. Those results suggested that kidney function is a main driver to control serum TMAO. In addition, authors used in vitro tissue culture and mouse model and confirmed that TMAO can contribute to kidney function decline. So the correlation between kidney function and serum fasting TMAO was well-investigated and explained. A lot of data were presented and the conclusion looks convincing. However, several concerns need be addressed.

Kidney function changes definitely affect a lot of serum metabolites’ concentrations since the metabolites are kept filtered and reabsorbed in kidney in each heart’s diastole and systole cycle. TMAO has been reported as uremic toxin, which is due to kidney function insufficiency. In Figure 1C, authors presented several urine metabolites in predication of serum TMAO variance, the urine metabolite concentrations normalized to creatinine before the association assay or authors used the absolute concentrations? In addition, how about the SHAP value of urine TMAO? If using other metabolites authors mentioned in Figure 1C as dependent variable as TMAO, how much percentage variance can be explained by kidney function?

In Fig 4B and supplementary Figure 6B, why the ERK1/2 intensity are different between MEK inhibitor, trametinib, or Ca²⁺ chelator, BAPTA-AM vs vehicle control? Is it due to sample loading amount different or trametinib, BAPTA-AM affecting ERK1/2 expression?

On the diet contributing to TMAO variance, authors in their manuscript stated that it plays a minor role. However, several clinical trials by strict experimental design indisputably confirmed that diet can contribute to a large variance of serum TMAO, which suggests that the diet questionnaires and its accompanying model “alternative Healthy Eating, Dietary Approaches to Stop Hypertension⁶⁴ (DASH) and Dietary Diversity⁶⁵ (DDS) scores” to evaluate diet nutrient has its limit, underestimating the diet effect.

In page 22, line#216, R²=0.009, is not consistent to that was shown in page 24 Fig 2, R²=0.0082.

Reviewer #2 (Remarks to the Author):

This is an interesting and informative paper. Circulating TMAO was found to be increased with cardiometabolic disease severity and associated with kidney function. With the treatment of reno-protective drugs, the authors detected lower circulating TMAO concentration and lower cardiovascular risk. However, the novelty is not enough, a meta-analysis reported that advanced CKD was associated with increase in TMAO concentration and subjects with high concentrations of TMAO had a decrease in glomerular filtration rate (PMID: 33751019). I have some concerns as follows.
1. Authors should be careful about using the word ‘real-life setting’. I believe authors tried to say their results should be applicable/close to daily clinical practice thus using ‘real-life setting’. However, the population they used were extracted from cohort studies which selected participants with multiple inclusion/exclusion criteria. This would, to some extent, hamper the generalizability of the results to all the target patients met in clinics. Therefore, I do not agree with emphasizing the current study as a

'real-life' study.

2. In line 166, Xgboost algorithm was applied together with five-fold cross-validation, whereas the datasets were not pre-split into training and testing parts. In this case, although five-fold cross-validation was used, the algorithm would see all the data while training the models, which declares optimization of the R2 and the risk of over-fitting. This should be mentioned in the method or discussion section.

3. In line 196, EV 21% versus 18% was considered significant, but the magnitude was small. Authors should mention in the methods that how much of the EV is good and how much of the improvement is relevant.

4. In the method section line 1010, Xgboost decision tree algorithm was used with the reason 'Xgboost consistently outperforms other algorithms in Kaggle competitions for tabular data'. Do authors mean Xgboost is always to be the best one in all the applicable situations? I doubt that, given the fact that existing literatures tend to compare multiple machine learning algorithms and then select the most suitable one. Authors should try other algorithms, either decision tree based or not, on the data to figure out whether Xgboost is the best one.

5. In line 1052, propensity score matching was done with age, sex, disease group and hypertension status as covariates. How did you select these variables? Why was BMI not included? BMI should be an important feature in T2D patients.

Reviewer #3 (Remarks to the Author):

1) The title (relationship between kidney function and TMAO) is not related to the study outcomes reported in the abstract (host variables contributing to fasting TMAO levels and therapeutic means);

2) The authors stated: "kidney function was the primary variable predicting circulating TMAO" – some parameters should be reported to highlight the statistical significance (area under the curve, p-value);

3) The authors should describe the novelty of the study, contrasting with already published studies and meta-analysis on large population sample size (e.g., <https://www.ncbi.nlm.nih.gov/pmc/articles/PMC9012260/>, <https://pubmed.ncbi.nlm.nih.gov/33751019/>, <https://doi.org/10.3390/toxins11110635>);

4) Cardiometabolic disease (and its severity) should be defined and described in the methods section;

5) The following statement should be included in the results section (or discussion), rather than in the introduction, as it provides insights into some of the study findings: "Interestingly, patients with T2D in the cohort prescribed new-generation anti-diabetics (GLP-1 Receptor Agonists; GLP-1RAs) with evidenced reno-protective effects had lower serum circulating TMAO levels when compared to propensity-score matched controls"

6) Outcomes of the study should be defined and described in the methods section;

7) It would be useful to discuss the possible mechanism of GLP-1Ras impact on TMAO levels and renal function in diabetic patients (e.g., gut microbiota composition modulation - <https://doi.org/10.3389/fendo.2021.814770>);

8) The authors stated: "In agreement with a recent study³, we did not find any significant association between habitual consumption of red meat and fasting serum levels of TMAO". However, there were other reports that documented an increase in TMAO levels, linked to an increased red meat intake (e.g., <https://pubmed.ncbi.nlm.nih.gov/30535398/>). Therefore, these discrepancies should be appropriately discussed.

REVIEWER COMMENTS

Reviewer comments are appended *verbatim* in blue, with author responses in black.

Reviewer #1 (Remarks to the Author):

In this manuscript, authors recruited 837 BMI-spectrum (BMIS) patients, 561 type II diabetes (T2D) patients and 337 patients with ischemic heart disease, and collected diet information and tissues to measure serum metabolites, urine metabolites and do metagenomics sequencing. Authors used “explainable” machine learning and tested each monitored parameter in predicting fasting plasma TMAO variance and found that age is the largest factor, then kidney function and its related parameters, and diet and microbiome’s contribution are very small. Authors also compared serum TMAO in T2D with prescribed new-generation anti-diabetics (GLP-1 Receptor Agonists; GLP-1RAs) with evidenced reno-protective effects vs without anti-diabetics in the collected cohort and found that the former group had significantly lower serum TMAO than the latter group. Those results suggested that kidney function is a main driver to control serum TMAO. In addition, authors used in vitro tissue culture and mouse model and confirmed that TMAO can contribute to kidney function decline. So the correlation between kidney function and serum fasting TMAO was well-investigated and explained. A lot of data were presented and the conclusion looks convincing. However, several concerns need be addressed.

We thank the reviewer for the overall positive evaluation of our work and for the reviewer’s suggestions that greatly improved the clarity of our manuscript.

Kidney function changes definitely affect a lot of serum metabolites’ concentrations since the metabolites are kept filtered and reabsorbed in kidney in each heart’s diastole and systole cycle. TMAO has been reported as uremic toxin, which is due to kidney function insufficiency. In Figure 1C, authors presented several urine metabolites in predication of serum TMAO variance, the urine metabolite concentrations normalized to creatinine before the association assay or authors used the absolute concentrations?

For the TMAO predictions using urine ¹H-NMR as input (**Figure 2A**) we did not correct for creatinine and have used in our calculations absolute concentrations derived from B.I QUANT-UR algorithm using the IVDr platform (Bruker, <https://www.bruker.com/en/products-and-solutions/mr/nmr-clinical-research-solutions/b-i-quant-ur.html>; please see Methods lines 902-907).

ACTION POINT:

We have now repeated the analysis using as input relative metabolite concentrations after normalising for urinary creatinine. This resulted in similar circulating TMAO predictions. We have included this new analysis in our revised manuscript as **Supplemental Figure 2A** (please see below).

Supplemental Figure 2A. Coefficients of determination (Explained Variance) of predicted circulating TMAO levels determined by xgboost algorithms after 5-fold cross-validation in the left-out group (Suppl.Table.4 for N numbers and optimized xgboost parameters), trained with input urinary metabolite absolute levels computed by the IVDr algorithm from $^1\text{H-NMR}$ spectra, excluding methylamines (TMA and dimethylamine) and corrected by urinary creatinine (also computed with the IVDr pipeline; Corrected) *versus* TMAO Explained Variance computed from the absolute levels of the same metabolites uncorrected for creatinine (Uncorrected).

We also added in the results section a sentence stating: lines 213-216 “ $^1\text{H-NMR}$ urine metabolomics, excluding TMA and dimethylamine, was the worst predictor explaining 1.5% of TMAO variance and correcting for urine creatinine, computed by $^1\text{H-NMR}$ (Methods), did not improve predictions explaining 1% of TMAO variance (**Suppl.Fig.2A**).”

In addition, how about the SHAP value of urine TMAO?

In our study we sought to identify variables most strongly predictive (and therefore most likely to affect) circulating TMAO levels. Consequently, we have removed urinary methylamines (TMA and dimethylamine) from our $^1\text{H-NMR}$ (and full) models. The IVDr platform does not provide urinary TMAO absolute quantifications, so urinary TMAO quantifications were not present as input in our models. Consequently, no SHAP value was computed for urinary TMAO.

ACTION POINT:

We now highlight this in the Results section (line 214).

If using other metabolites authors mentioned in Figure 1C as dependent variable as TMAO, how much percentage variance can be explained by kidney function?

We thank the reviewer for this suggestion.

ACTION POINT:

We have now built confirmatory linear regression models, as the reviewer suggests, using the urine (betaine_U, oxaloacetate_U) and plasma (butyryl-carnitine, betaine, *p*-cresol) metabolites as dependent variables that were most predictive of TMAO in our ML analysis (**Figure 2C**) with eGFR as the independent variable. We compared those to a similar model with TMAO as the independent variable. We observed that eGFR explained 7% of serum TMAO variance (**Figure 2E**) whilst the explained variance of the other metabolites ranged from 6% to 1.4%. This analysis is now included in our revised manuscript as **Supplemental Figure 4A-E**.

Additionally, to assess the relative importance of circulating metabolites in predicting eGFR in BMIS, we used boosted trees similarly to TMAO with circulating metabolites as input variables. Circulating metabolites explained 25% of eGFR variance in BMIS and SHAP analysis revealed that TMAO was the top metabolite of bacterial origin and one of the top 4 metabolites contributing to eGFR predictions (**Supplemental Figure 4F**), supporting our hypothesis that circulating TMAO levels are strongly interlinked to kidney function. Please see the new Supplemental Figure below.

Collectively, this analysis highlights that kidney clearance impacts on fasting circulating metabolite levels to a different extent, contrary to what would have been observed if it was affecting all metabolites similarly. We make now this point in the results section.

Specifically, we state: lines 250-262 “To determine how much of the variances of TMAO and of the other metabolites most strongly associated with its levels in our ML models (**Figure 2C**; butyryl-carnitine, betaine, *p*-cresol and betaine_U, oxaloacetate_U) is explained by eGFR we built linear regression models with each metabolite as the dependent variable and eGFR as the independent variable. Kidney function explained 7% of TMAO variance in BMIS (**Figure 2F**; Pearson’s $r=-0.26$, $P=5.4 \times 10^{-14}$, $N=837$) whilst the explained variance for the other metabolites ranged from 6% to 1.4% for *p*-cresol and urinary oxaloacetic acid, respectively (**Suppl.Fig.4A-E**). To further assess the varying relationship of metabolites with eGFR we computed boosted trees models predicting eGFR with serum metabolomics as the input variable, with a similar methodology to TMAO. Serum metabolomics predicted on average 25% of eGFR variance after 100 iterations. From all the metabolites in our analysis TMAO was the top microbiota-derived compound that affected most strongly eGFR predictions (**Suppl.Fig.4F**) in line with its reported exclusive glomerular secretion³². Collectively this analysis suggests that in our population, TMAO is strongly interlinked with eGFR.”

Supplemental Figure 4. TMAO is the measured metabolite most strongly associated with eGFR in BMIS (N=767). Linear-regression-based scatterplot showing correlation between metabolites most strongly predictive of TMAO (**Figure 2C**; serum p-cresol (**A**), serum betaine (**B**), serum butyryl-carnitine (**C**), urinary betaine (**D**), urinary oxaloacetic acid (**E**)) as dependent variables and estimated Glomerular Filtration Rate (eGFR, ml/min/1.73m²) as independent variable. Insert, explained variance (R²). (**F**) Swarm plots of SHAP values (impact on eGFR model predictions) for each BMIS MetaCardis participant with available serum metabolomics (N=767); represented by individual dots, for the top 20 metabolites contributing to eGFR predictions, computed from xgboost algorithms trained on serum

metabolomics. Numbers denote mean absolute SHAP values from all BMIS participants (in descending order) next to their corresponding metabolite. Dots are colored by the inverse-normalized value of their corresponding metabolite. See **Suppl.Table.4** for N numbers and optimized xgboost parameters.

In Fig 4B and supplementary Figure 6B, why the ERK1/2 intensity are different between MEK inhibitor, trametinib, or Ca²⁺ chelator, BAPTA-AM vs vehicle control? Is it due to sample loading amount different or trametinib, BAPTA-AM affecting ERK1/2 expression?

The reviewer is correct, not the same amount of protein was loaded in different western blots opting instead for maximum protein loading, as calculated by BCA assays (see **Methods**). Within each western blot, the same amount of protein was loaded between samples and normalised accordingly.

For the specific experiments the reviewer highlights (trametinib and BAPTA-AM) these are short-term experiments (maximum duration up to 30min), therefore changes in ERK1/2 expression are not expected. Any apparent differences in expression of ERK1/2 are most probably due to technical artifacts from stripping the membrane of the pERK1/2 antibody for subsequent re-probing with the ERK1/2 antibody. Since we made comparisons within the same experiment, this does not affect our conclusions.

On the diet contributing to TMAO variance, authors in their manuscript stated that it plays a minor role. However, several clinical trials by strict experimental design indisputably confirmed that diet can contribute to a large variance of serum TMAO,

The reviewer makes an important point. We agree with the reviewer that the source of TMAO is dietary as has been shown by several preclinical models and human interventional trials that we also quote (references 8, 9, 10 in the introduction of our manuscript). This was originally highlighted in the introduction and in the discussion but not in the results section:

Lines (453-463): “With a few notable exceptions³, the contribution of diet and in particular red meat and *L*-carnitine to TMAO levels, showing an increase following intervention, has predominately been examined in metabolically healthy volunteers⁹⁻¹¹. In such interventions, often *L*-carnitine has been provided as a dietary supplement, which is poorly absorbed in the small intestine (~12%), as opposed to dietary *L*-carnitine (~71%)⁵³, and therefore may be more available for microbial catabolism in the upper gut and large intestine, leading to overestimating its role in TMA and, thereby, TMAO production. Our observations, similar to the report by Li and colleagues³, suggests that in non-interventional settings where individuals habitually consume meat (75 to 233 g/day in European adults³⁷), this contributes minimally to fasting circulating TMAO variability, possibly limiting the isolated effect of dietary manipulation on TMAO levels in non-interventional settings, aside strict vegans or vegetarians.”

Therefore, we respectfully submit that our analyses do not contradict the dominant role of diet in TMAO production. Instead, our data indicate that in non-interventional settings, where individuals (aside strict vegans or vegetarians; that are infrequent in MetaCardis with 65/1847 participants reporting no red meat consumption) daily consume meat (75 – 233 g / day for European adults) the most important determinant of fasting circulating levels of TMAO (and consequently associated excess cardiovascular risk) is not its production from dietary sources (or the microbiome composition) but its clearance by the kidney.

We acknowledge that making this crucial point at the end of our manuscript in the discussion can be a source of confusion for the reader.

ACTION POINT:

We now make this distinction early in the results section where the dietary analyses are presented. Specifically, we now state:

Line 281 – 291: “Our findings that diet and particularly meat consumption does not associate with increased TMAO levels in our population does not contradict a number of well-designed human interventional trials^{9–11} that have established a clear link between meat intake or *L*-carnitine supplementation and TMAO circulating levels. Instead, collectively, our analyses suggest that in non-interventional settings where most individuals consume meat daily (75 to 233 g/day for European adults³⁷), clearance by the kidney and not dietary intake of TMAO precursors is the major determinant of fasting circulating TMAO levels and therefore of the excess cardiovascular risk associated with elevated TMAO (vegetarians and strict vegans aside, who are infrequent in the MetaCardis cohort with only 65/1741 participants reporting no red meat consumption).”

which suggests that the diet questionnaires and its accompanying model “alternative Healthy Eating, Dietary Approaches to Stop Hypertension⁶⁴ (DASH) and Dietary Diversity⁶⁵ (DDS) scores” to evaluate diet nutrient has its limit, underestimating the diet effect.

Regarding the validity of our dietary parameters (DASH and DDS). These were derived from annual Food Frequency Questionnaires (FFQs), according to formulas referenced in the Methods (references 64 and 65). Annual FFQs provide a smoothed dietary estimate, suitable for nutritional exposure in a context of chronic non-communicable diseases. MetaCardis FFQs were benchmarked for each country and validated by three web-based patient dietary recalls for a subset of participants (N=324) following standard procedures (for details please see reference 34 in our revised manuscript Verger *et al.* (2017). Dietary Assessment in the MetaCardis Study: Development and Relative Validity of an Online Food Frequency Questionnaire. *J Acad Nutr Diet* 117:878-888. doi: 10.1016/j.jand.2016.10.030).

Consequently, we submit that despite the inherent limitations of FFQs these were validated according to established methods and combined with the statistical power of MetaCardis reflect as faithfully as possible long-term dietary patterns in our study. Moreover, our observation regarding the statistically non-significant relationship between fasting circulating TMAO and habitual red meat intake in non-interventional settings is in agreement with the recent study of Li and colleagues (reference 3 in our manuscript).

ACTION POINT:

We now highlight that the MetaCardis FFQs were validated by patient recall in the Methods (lines 895-897).

In page 22, line#216, R2=0.009, is not consistent to that was shown in page 24 Fig 2, R2=0.0082.

We thank the Reviewer for spotting this, we apologize for this error, which has now been rectified (please see line 297 in our revised manuscript).

Reviewer #2 (Remarks to the Author):

This is an interesting and informative paper. Circulating TMAO was found to be increased with cardiometabolic disease severity and associated with kidney function. With the treatment of reno-protective drugs, the authors detected lower circulating TMAO concentration and lower cardiovascular risk.

We thank the reviewer for finding our study of interest and informative and for the reviewer's suggestions that greatly enhanced the clarity of our work.

However, the novelty is not enough, a meta-analysis reported that advanced CKD was associated with increase in TMAO concentration and subjects with high concentrations of TMAO had a decrease in glomerular filtration rate (PMID: 33751019). I have some concerns as follows.

We never claimed to be the first to discover an inverse association between TMAO and kidney function. We have referenced the first studies that made this connection (References 10, 19, 20) and we have now included in our revised manuscript the informative meta-analysis that the reviewer highlighted (Reference 33). However, these reports predominately refer to advanced CKD, which leaves unaddressed gaps regarding the preclinical stages of CKD typically in a cardiometabolic context.

The novelty of our study is to objectively investigate contributions to circulating TMAO levels in a non-CKD population with a range of non-clinical kidney function across the cardiometabolic disease spectrum. Our integrative approach allowed us then to uncover novel biological insight (experimental causality experiments confirming kidney scarring and the 2-hit model of TMAO action on the kidney) and a potentially clinically actionable intervention to reduce TMAO (GLP-1RAs). This has never been shown before.

Please see also our response to reviewer's 3 similar query below.

ACTION POINT:

We now make this crucial point early in our revised in the Results section.

Specifically in lines 263-268 we now state:

“Several reports have previously highlighted the inverse correlation between TMAO and kidney function mostly in patients with Chronic Kidney Disease (CKD)^{10,19,20,32-34}, but there is limited evidence for the predominance of this relationship in the non-clinical range. The novelty of the present study includes ranking of a multitude of factors contributing to circulating TMAO levels and identification of kidney function as the top modifiable factor in a non-CKD population across the cardiometabolic disease spectrum.”

1. Authors should be careful about using the word 'real-life setting'. I believe authors tried to say their results should be applicable/close to daily clinical practice thus using 'real-life setting'. However, the population they used were extracted from cohort studies which selected participants with multiple inclusion/exclusion criteria. This would, to some extent, hamper the generalizability of the results to all the target patients met in clinics. Therefore, I do not agree with emphasizing the current study as a 'real-life' study.

We thank the reviewer for highlighting this. It appears our use of the phrase “real-life setting” in our manuscript can create confusion and mislead the reader.

ACTION POINT:

We now have changed “real-life setting” to “non-interventional setting” throughout our manuscript. This phrasing more clearly conveys the distinction between our study where participants chose their diet freely and dietary interventional studies.

2. In line 166, Xgboost algorithm was applied together with five-fold cross-validation, whereas the datasets were not pre-split into training and testing parts. In this case, although five-fold cross-validation was used, the algorithm would see all the data while training the models, which declares optimization of the R² and the risk of over-fitting. This should be mentioned in the method or discussion section.

We thank the reviewer for this suggestion. As described by the Reviewer, we performed our model validation following the procedure outlined in Bar *et al.* (2020) Nature 588:135–140 (Reference 23 in our revised manuscript), using 5-fold cross-validations with the random 80/20 partitioning resampled 100 times. The R² we report is the R² value obtained on the internal test set, which is less optimistic than the R² value obtained from the training set (see **Methods** in the original submission, lines 976-978 in the revised manuscript “For each round, we calculated the coefficient of determination using the rsq function from yardstick (v0.0.7) and the predicted regularized TMAO values.”). Although the risk of overfitting has been mitigated by this procedure, it is still possible that there remains residual overfitting in the reported values and we now acknowledge it in the **Methods**:

The MetaCardis study, is by design, a heterogeneous population: we recruited participants across the cardiometabolic disease spectrum resulting in co-morbidity and polypharmacy thus making the generation of equivalent training and test sets challenging. In this context, we opted instead for a model validation strategy using 5-fold cross-validation (train on 4/5 of our population, and then predict on the remaining 1/5) within our broad disease groups (BMIS, T2D and IHD) using all the participants and generating ensemble models by re-iterating the partitioning and 5-fold cross-validation a 100 times.

We agree with the reviewer that using 5-fold cross-validation within all disease group participants could potentially lead to overfitting. To minimize this risk, when building our models we took four steps: 1) during the model parameter optimization step, training was stopped if predictions were not improved for 10 rounds (early_stopping_rounds = 10). 2) We incorporated the regularization parameters lambda and gamma in our models, thus making them more conservative (Bar *et al.*, (2020) Nature 588:135-140). 3) We intentionally introduced randomness by using 0.8-0.9 of the available variables in each variable group (colsample_bytree=0.8-0.9 depending on parameter optimization; **Supplemental Tables 4-6**) for the training of each tree, as a way to minimize overfitting. 4) all our conclusions are based on ensemble models (the average of 100 independent runs) that in combination with the introduced randomness and the out-of-sample predictions (5-fold cross-validation) makes our models conservative. Please see also the XGBoost documentation (<https://xgboost.readthedocs.io/en/latest/index.html>)

ACTION POINT:

We now elaborate on each of these points in the Methods section of the revised manuscript. Please see lines 980-993 where we state:

“To minimize the risk overfitting, we took four steps:

- 1) during the model parameter optimization step, training was stopped if predictions were not improved for 10 rounds (`early_stopping_rounds = 10`).
- 2) We incorporated the regularization parameters λ and γ in our models, thus making our models more conservative²³.
- 3) We introduced randomness by using 0.8-0.9 of the available variables in each variable group (`colsample_bytree=0.8-0.9` depending on parameter optimization; **Supplemental Tables 4-6** for specific model parameters) for the training of each tree, as a way to minimize overfitting.
- 4) all our conclusions are based on ensemble models (the average of 100 independent runs) that in combination with the introduced randomness and the out-of-sample predictions (5-fold cross-validation) makes our models conservative.

Please see also the XGBoost documentation for additional information on model parameters (<https://xgboost.readthedocs.io/en/latest/index.html>)”.

3. In line 196, EV 21% versus 18% was considered significant, but the magnitude was small. Authors should mention in the methods that how much of the EV is good and how much of the improvement is relevant.

The gain in explained variance in **Figure 2E** between models trained to predict TMAO using all variables and the “top-SHAP” variables is statistically significant ($P < 2.2 \times 10^{-16}$, Mann-Whitney test). We have now included these P values in the respective figures where we compare these models (**Figure 2E**, **Figure 4C** & **Supplemental Figure 6C**) for clarity. We also now state specifically “statistically significant” in our revised manuscript to avoid confusion regarding the interpretation of “significant”. As stated in our manuscript, this improvement in prediction is indicative that the top features identified by SHAP are relevant to circulating TMAO and selecting only those reduces model noise thus improving predictions.

Defining how much EV “is good and how much of the improvement is relevant” is challenging, since, to the best of our knowledge, no established benchmarks are available, let alone universally accepted, for complex clinical systems. For comparison purposes, Bar *et al.* (Reference 23 in our revised manuscript, whose method we adapted to predict TMAO in our population) achieved EV of 14% for TMAO in their population which did not include kidney parameters (N=491; Supplementary table 6 line 314 in their publication).

Therefore, we believe the results derived from our models represent an improvement (even though of a modest magnitude) to comparable non-interventional studies and consequently are appropriate for computing the relative impact of variables on fasting TMAO levels.

4. In the method section line 1010, Xgboost decision tree algorithm was used with the reason ‘Xgboost consistently outperforms other algorithms in Kaggle competitions for tabular data’. Do authors mean Xgboost is always to be the best one in all the applicable situations? I doubt that, given the fact that

existing literatures tend to compare multiple machine learning algorithms and then select the most suitable one. Authors should try other algorithms, either decision tree based or not, on the data to figure out whether Xgboost is the best one.

We thank the reviewer for highlighting this. Our reasoning for using a boosted trees method was that a similar approach has recently successfully resolved the relationship between circulating metabolites and variable groups in a deeply-phenotyped human population, similar to ours (Bar *et al.* (2020) Nature 588:135–140).

ACTION POINT:

To confirm the Bar *et al.*²³ statement that a tree-based model performs better than conventional linear regression models we built least absolute shrinkage and selection operator (LASSO) models to predict circulating TMAO in BMIS (N=582), using again 5-fold cross validation with all the available variables (full model) as input. LASSO explained 14.9% of circulating TMAO variance, thus performing worse than our boosted trees models explaining 18.4 % of TMAO in BMIS (**Figure 2A**). We have now included this additional analysis as (**Suppl.Fig.3A**; please see below) and added a new methods section for the LASSO analysis (lines 1007-1017) in our revised manuscript.

We also state in the results section: lines 243-249 “To confirm that tree-based ML models are the most appropriate for our analysis we also built Least Absolute Shrinkage and Selection Operator (LASSO) models to predict circulating TMAO in the left-out group using again 5-fold cross-validation with all the available variables (full model) as input in BMIS (N=582; Methods). LASSO explained on average 14.9% of circulating TMAO variance after 100 iterations (**Suppl.Fig.3A**; see source data for lambda and R² values of each iteration) as opposed to 18.4% by boosted trees for the full model (**Figure 2A**). This analysis supports the appropriateness of tree-based ML models for predicting circulating TMAO in our population.”

Supplemental Figure 3. LASSO linear-regression models explain less circulating TMAO variance in BMIS (N=582) than boosted trees models. Explained Variance of predicted circulating TMAO levels determined by LASSO linear regression models in BMIS (N=582) after 5-fold cross-validation in the left-out group for 100 iterations. Explained variance from boosted trees models (Xgboost) from BMIS full model computed in **Figure 2A** are also included for comparison.

5. In line 1052, propensity score matching was done with age, sex, disease group and hypertension status as covariates. How did you select these variables? Why was BMI not included? BMI should be an important feature in T2D patients.

Propensity-score matching typically matches for variables strongly associated with and affect the clinical outcome of interest, *i.e.* kidney function in our study (associated with age, sex, hypertension, disease status). We performed this matching exclusively in MetaCardis patients with T2D, since those were the ones prescribed with GLP-1RAs. After this matching strategy we did not observe any significant differences for BMI ($P=0.28$), Age ($P=0.98$), Glycated Haemoglobin (Hba1c (%), $P=0.3$), Systolic Blood Pressure ($P=0.51$), or drugs intake (other than GLP-1RAs, which is also reflected in the number of anti-diabetic therapies prescribed) for the two groups (please see **Supplemental Figure 10** and **Supplemental Table 7**). Hence, we opted not to additionally match for BMI since this was already not significantly different between our groups.

Reviewer #3 (Remarks to the Author):

1) The title (relationship between kidney function and TMAO) is not related to the study outcomes reported in the abstract (host variables contributing to fasting TMAO levels and therapeutic means);

We thank the reviewer for this suggestion.

ACTION POINT:

We have now shortened the abstract in accordance to Nature Communications guidelines (maximum 150 words). To better align the abstract with our article title we now state:

Lines 85-88. “Our analyses uncovered a bidirectional relationship between kidney function and TMAO that can potentially be modified by reno-protective anti-diabetic drugs and suggest a clinically actionable intervention for decreasing TMAO-associated excess cardiovascular risk.”

We have also now modified the title of our study to:

“Evidence of a causal and modifiable relationship between kidney function and circulating trimethylamine *N*-oxide.”

This is in accordance to Nature Communications guidelines stipulating 15 words limit for article titles.

2) The authors stated: “kidney function was the primary variable predicting circulating TMAO” – some parameters should be reported to highlight the statistical significance (area under the curve, p-value);

ACTION POINT:

In response also to a comment from reviewer 1 we have now added a linear regression model between eGFR and circulating TMAO in **Figure 2F** in our manuscript (please see below). In BMIS (N=862), eGFR inversely associates with circulating TMAO (Pearson’s $r = -0.26$, $P=5.4 \times 10^{-14}$) and explains 7% of its variance.

Figure 2F. Linear-regression-based scatterplot showing correlation between serum TMAO (log-transformed for visualization purposes) and estimated Glomerular Filtration Rate (eGFR, ml/min/1.73 m²). Insert; unadjusted Pearson’s r , P value and explained variance (R^2).

Please also see lines 253-4 in our revised manuscript where we now state: “Kidney function explained 7% of TMAO variance in BMIS (**Figure 2F**; Pearson’s $r=-0.26$, $P=5.4 \times 10^{-14}$, $N=837$)”.

3) The authors should describe the novelty of the study, contrasting with already published studies and meta-analysis on large population sample size (e.g., <https://www.ncbi.nlm.nih.gov/pmc/articles/PMC9012260/>, <https://pubmed.ncbi.nlm.nih.gov/33751019/>, <https://doi.org/10.3390/toxins11110635>);

We thank the reviewer for the opportunity to highlight the novelty of our work.

We do not claim to be the first to report a strong inverse association between kidney function and circulating TMAO. Indeed, we reference a number of studies that have already done so in the introduction (References 10, 19, 20) and in our revised submission we now include the helpful references that the reviewer highlighted along with an additional meta-analysis suggested by reviewer 2. However, we note that these studies predominately refer to advanced CKD and leave unaddressed gaps regarding prodromal stages of CKD in the context of cardiometabolic co-morbidities.

With that in mind, ours is the first report objectively ranking variables associated with fasting circulating TMAO in a population across the cardiometabolic disease spectrum. This approach led us to uncover novel biological insight (Ca²⁺-dependant activation of the ERK1/2 pathway by TMAO and enhanced myofibroblast differentiation in conjunction with TGF-β1 in primary renal fibroblasts coupled with worse kidney scarring in a rodent pure fibrosis model in the absence of other co-morbidities leading us to propose the 2-hit model of TMAO action on the kidney) and a potentially clinically actionable intervention to reduce TMAO (GLP-1RAs). This has never been shown before.

Please see also our response to reviewer's 2 similar query above.

ACTION POINT:

We highlight now this point in relation to kidney function early in the Result section of our revised manuscript.

lines 263-268: "Several reports have previously highlighted the inverse correlation between TMAO and kidney function predominately in patients with Chronic Kidney Disease (CKD)^{10,19,20,32-34}, but there is limited evidence for this relationship in the non-clinical range. The novelty of the present study includes ranking of a multitude of factors contributing to circulating TMAO levels and identification of kidney function as the top modifiable factor in a non-CKD population across the cardiometabolic disease spectrum."

4) Cardiometabolic disease (and its severity) should be defined and described in the methods section;

In brief, cardiometabolic disease is used as an umbrella term for cardiovascular risk factors and overt disease with a clear focus on the progression from obesity toward overt T2D and cardiovascular phenotypes including ischemic heart disease and heart failure. By "severity" we meant the spectrum from health to dysmetabolism (obesity and T2D) and finally to overt IHD. These cardiovascular risk factors and diseases were defined according to internationally recognised clinical guidelines written by international associations/ working groups of disease.

ACTION POINT:

We have now expanded the cohort description section in the Methods to accommodate this information. This section reads now as follows:

Lines: 867-884: “Briefly, MetaCardis is a cross-sectional study that recruited individuals at increasing stages of dysmetabolism and IHD severity (ranging from metabolically healthy, metabolic syndrome and/or obesity, T2D, IHD), aged 18–75 years old and recruited from Denmark, France and Germany between 2013 and 2015. The overarching goal of the trial was to investigate the impact of qualitative and quantitative changes in the gut microbiota on the pathogenesis of cardiometabolic diseases (CMDs) and their associated co-morbidities (ClinicalTrials.gov Identifier: NCT02059538). For the present study, patients were subclassified in three groups: BMI-spectrum patients (BMIS²⁶; N=837), encompassing MetaCardis participants presenting with metabolic syndrome-related risk factors or conditions (hypertension, as defined by the American Heart Association⁶⁴; obesity, as defined by the World Health Organization⁶⁵ and metabolic syndrome, as defined by the International Diabetes Federation⁶⁶) and patients diagnosed with type-2 diabetes (T2D, as defined by the American Diabetes Association⁶⁷; N=561) or ischaemic heart disease (IHD; N=356). The IHD group comprised patients with Acute (<15days) Coronary Syndrome (ACS; N=106), Chronic IHD (CIHD; N=157) with normal Left Ventricular Ejection Fraction (LVEF) determined by echocardiography and Heart Failure patients (HF; N=93, LVEF<45%). Cardiometabolic disease is used as an umbrella term for all the above cases and severity of cardiometabolic disease refers, in this manuscript, to the progression from single risk factors such as obesity to overt T2D and cardiac phenotype (ischemic heart disease and heart failure).”

5) The following statement should be included in the results section (or discussion), rather than in the introduction, as it provides insights into some of the study findings: “Interestingly, patients with T2D in the cohort prescribed new-generation anti-diabetics (GLP-1 Receptor Agonists; GLP-1RAs) with evidenced reno-protective effects had lower serum circulating TMAO levels when compared to propensity-score matched controls”

This statement has now been modified, in line with summarizing the main findings of our study in the last paragraph of the introduction.

ACTION POINT:

Specifically in lines 176-179 we now state: “Further supporting the strong interplay between kidney function and fasting circulating TMAO, patients with T2D in the cohort prescribed new-generation anti-diabetics (GLP-1 Receptor Agonists²⁴; GLP-1RAs) with evidenced reno-protective effects²⁵ had lower serum circulating TMAO levels when compared to propensity-score matched controls (**Figure 1**).”

6) Outcomes of the study should be defined and described in the methods section;

MetaCardis is a cross-sectional study, where patients in the cardiometabolic disease spectrum (ranging from metabolically healthy to heart failure) have been recruited and phenotypically assessed at one point in time. The overarching goal of the trial was to investigate the impact of qualitative and quantitative changes in the gut microbiota on the pathogenesis of cardiometabolic diseases (CMDs) and their associated co-morbidities (please also see the description in NCT clinical trial (ClinicalTrials.gov Identifier: NCT02059538). Therefore, there are no clinical outcomes to report.

ACTION POINT:

For the avoidance of any confusion, we now state that explicitly in the Methods section (lines 867-873). “Briefly, MetaCardis is a cross-sectional study that recruited individuals at increasing stages of dysmetabolism and IHD severity (ranging from metabolically healthy, metabolic syndrome and/or obesity, T2D, IHD), aged 18–75 years old and recruited from Denmark, France and Germany between 2013 and 2015. The overarching goal of the trial was to investigate the impact of qualitative and quantitative changes in the gut microbiota on the pathogenesis of cardiometabolic diseases (CMDs) and their associated co-morbidities (ClinicalTrials.gov Identifier: NCT02059538).”

7) It would be useful to discuss the possible mechanism of GLP-1Ras impact on TMAO levels and renal function in diabetic patients (e.g., gut microbiota composition modulation - <https://doi.org/10.3389/fendo.2021.814770>);

We thank the reviewer for this informative reference.

ACTION POINT:

We have now included in our manuscript a brief comment regarding potential mechanism(s) of GLP-1RAs action in lines 501-503 of our revised manuscript, where we state: “Differences in microbiota composition were predictive of glycemic responses to GLP-1RA intake⁶¹ and further work is required to determine factors influencing GLP-1RA-mediated reno-protection which appear to be independent of improvements in glycemic control⁶².”

It should be noted though that the reno-protective effect of GLP-1RAs (similar to SGLT2i) is independent of improvements in glycaemic control. Please see reference 62 in our revised manuscript; Tuttle, K. R. et al. Dulaglutide versus insulin glargine in patients with type 2 diabetes and moderate-to-severe chronic kidney disease (AWARD-7): a multicentre, open-label, randomised trial. *Lancet Diabetes Endocrinol.* 6, 605–617 (2018). Additionally, in the propensity score matched patients with diabetes medicated with GLP-1RAs there is no significant difference in glycated haemoglobin between groups (**Supplemental Figure 10D**) further suggesting a reno-protective action of these drugs uncoupled from benefits on glucose handling.

8) The authors stated: “In agreement with a recent study³, we did not find any significant association between habitual consumption of red meat and fasting serum levels of TMAO”. However, there were other reports that documented an increase in TMAO levels, linked to an increased red meat intake (e.g., <https://pubmed.ncbi.nlm.nih.gov/30535398/>). Therefore, these discrepancies should be appropriately discussed.

We thank the reviewer for giving us the opportunity to clarify this important point.

Similar to our response to a point made by reviewer 1 (please see above), we agree that the source of TMAO is dietary as demonstrated in a number of human interventional trials that we also reference (references 9-11 in the introduction of our manuscript, including the study the reviewer highlights). However, as we emphasize in the discussion:

Lines (453-463): “With a few notable exceptions³, the contribution of diet and in particular red meat and *L*-carnitine to TMAO levels, showing an increase following intervention, has predominately been examined in metabolically healthy volunteers⁹⁻¹¹. In such interventions, often *L*-carnitine has been provided as a dietary supplement, which is poorly absorbed in the small intestine (~12%), as opposed

to dietary *L*-carnitine (~71%)⁵³, and therefore may be more available for microbial catabolism in the upper gut and large intestine, leading to overestimating its role in TMA and, thereby, TMAO production. Our observations, similar to the report by Li and colleagues³, suggests that in non-interventional settings where individuals habitually consume meat (75 to 233 g/day in European adults³⁷), this contributes minimally to fasting circulating TMAO variability, possibly limiting the isolated effect of dietary manipulation on TMAO levels in non-interventional settings, aside strict vegans or vegetarians.”

Therefore, we submit that our analyses do not contradicting the dominant role of diet as a source of TMAO production. Instead, our work suggests that in non-interventional settings, where individuals (aside strict vegans or vegetarians, that are infrequent (65/1741 report no red meat consumption) in MetaCardis) daily consume meat (75 – 233 g / day for European adults) the most important determinant of fasting circulating levels of TMAO (and consequently associated excess cardiovascular risk) is not its production from dietary sources (or the microbiome composition) but its clearance by the kidney.

ACTION POINT:

We acknowledge that we need to make this crucial distinction early in order to avoid any confusion to the reader. We, therefore, now state in the results section where the dietary analyses are first presented:

Lines 281 – 291: “Our findings that diet and particularly meat consumption does not associate with increased TMAO levels in our population does not contradict a number of well-designed human interventional trials⁹⁻¹¹ that have established a clear link between meat intake or *L*-carnitine supplementation and TMAO circulating levels. Instead, collectively, our analyses suggest that in non-interventional settings where most individuals consume meat daily (75 to 233 g/day for European adults³⁷), clearance by the kidney and not dietary intake of TMAO precursors is the major determinant of fasting circulating TMAO levels and therefore of the excess cardiovascular risk associated with elevated TMAO (vegetarians and strict vegans aside, who are infrequent in the MetaCardis cohort with only 65/1741 participants reporting no red meat consumption).”

REVIEWERS' COMMENTS

Reviewer #1 (Remarks to the Author):

Authors responded to my concerns and modified the manuscript accordingly. I have no more concerns.

Reviewer #2 (Remarks to the Author):

The author carefully revised the manuscript and answered all my questions.

Reviewer #3 (Remarks to the Author):

I appreciate the quality of the study and the robust design methodology. The statistical analysis section is well written, with a meticulous description of the approach towards biomarker selection and modelling. The manuscript is a well conducted analysis of that could offer some interesting insights.